# FeatInv: Spatially resolved mapping from feature space to input space using conditional diffusion models

**Nils Neukirch**  *nils.neukirch@uol.de*
*Division AI4Health*
*Carl von Ossietzky Universität Oldenburg*

**Johanna Vielhaben**  *johanna.vielhaben@hhi.fraunhofer.de*
*Explainable Artificial Intelligence Group*
*Fraunhofer Heinrich-Hertz-Institute*

**Nils Strodthoff**  *nils.strodthoff@uol.de*
*Division AI4Health*
*Carl von Ossietzky Universität Oldenburg*

**Reviewed on OpenReview:** *https://openreview.net/forum?id=UtE1YnPNgZ*

## Abstract

Internal representations are crucial for understanding deep neural networks, such as their properties and reasoning patterns, but remain difficult to interpret. While mapping from feature space to input space aids in interpreting the former, existing approaches often rely on crude approximations. We propose using a conditional diffusion model - a pretrained high-fidelity diffusion model conditioned on spatially resolved feature maps - to learn such a mapping in a probabilistic manner. We demonstrate the feasibility of this approach across various pretrained image classifiers from CNNs to ViTs, showing excellent reconstruction capabilities. Through qualitative comparisons and robustness analysis, we validate our method and showcase possible applications, such as the visualization of concept steering in input space or investigations of the composite nature of the feature space. This approach has broad potential for improving feature space understanding in computer vision models.

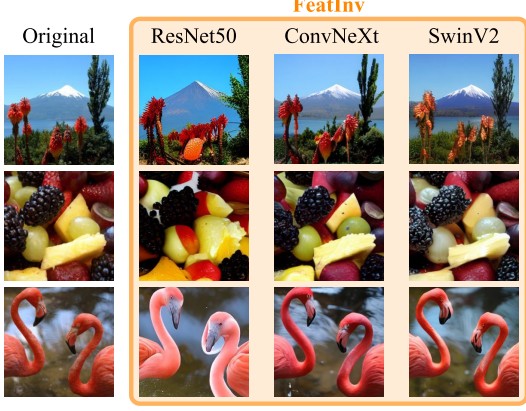

Figure 1: *FeatInv* learns a probabilistic mapping from feature space to input space and thereby provides a visualization of how a sample is perceived by the respective model. The goal is to identify input samples within the set of natural images whose feature representations align most closely with the original feature representation of a given model. In this figure, we visualize reconstructed samples obtained by conditioning on the feature maps of the penultimate layer from ResNet50, ConvNeXt and SwinV2 models.

# 1 Introduction

The feature space is vital for understanding neural network decision processes as it offers insights into the internal representations formed by these models as they process input data. While it serves as the foundation for many modern explainability approaches (Rai et al., 2024; Bereska & Gavves, 2024), its importance extends beyond interpretability. The feature space provides a rich resource for investigating fundamental properties of deep neural networks, including their robustness against perturbations, invariance characteristics, and symmetry properties (Bordes et al., 2022). By analyzing the geometry and topology of these learned representations, researchers can gain insights into model generalization capabilities, failure modes, and the emergence of higher-order patterns in the data. This perspective enables advancements in theoretical understanding of neural networks while informing practical improvements in architecture design and training methodologies.

An important challenge in examining the feature space is establishing a connection back to the input domain, especially for classification models that map to labels rather than the same domain as the input. One aspect of this challenge involves identifying which part of the input a particular region or unit in feature space is sensitive to. GradCAM (Selvaraju et al., 2017) pioneered this by linearly upsampling a region of interest in feature space to the input size. However, linear upsampling imposes a rather strong implicit assumption. As an alternative, one might consider the entire receptive field of a feature map location, yet in deep architectures these fields tend to be broad and less informative.

The more intricate second aspect of this challenge is to derive a mapping from the entirety of the feature space representation back to the input domain – beyond mere localization. Recent works proposed to leverage conditional generative models to learn such a mapping by conditioning them on feature maps (Bordes et al., 2022; Dosovitskiy & Brox, 2016; Rombach et al., 2020). However, these approaches either build on pooled feature maps (discarding finegrained spatial details of the feature map), only provide deterministic mappings (overlooking the inherent uncertainty of this ill-posed problem), or do not utilize state-of-the-art generative models. Related approaches such as diffusion autoencoders (Preechakul et al., 2022) show that diffusion models can indeed be fitted with meaningful and decodable latent representations that enable near-exact reconstructions and the manipulation of semantic attributes. However, their latent codes are global and not spatial, whereas our focus is on conditioning spatially resolved feature maps to preserve the fine-grained structure. We argue that a lack of faithfulness, i.e., the inability to reconstruct spatially resolved feature maps, is detrimental for downstream applications, e.g., in explainable AI or for robustness analyses, as it fails to reflect the true relation between feature space and input space in the model. In addition, pooled representations are not meaningful for the purpose of studying feature representations of intermediate layers. To the best of our knowledge, there is no probabilistic model that provides high-fidelity input samples when conditioned on a spatially resolved feature map – thereby integrating both aspects of the challenge described above. We aim to close this gap with this submission.

More specifically, in this work we put forward the following contributions:

1. We demonstrate the feasibility of learning high-fidelity mappings from a spatially resolved feature space to input space using a conditional diffusion model of the ControlNet-flavor, as exemplified in Fig. 1. We investigate this for different computer vision models, ranging from CNNs to ViTs.

2. We provide quantitative evidence that generated samples align with the feature maps of the original samples and that the samples represent high-fidelity natural images, see Tab. 1. and carry out a qualitative model comparison, see Fig. 3.

3. We provide robustness analysis, see Tab. 2 as well as further analysis demonstrating the generalizability of our approach across different ConvNeXt stages (see Fig. 4), its behavior on out-of-distribution data (see Fig. 5), as well as its robustness to corrupted images (see Sec. 4.2).

4. We provide a specific use-cases for the application of the proposed methodology to visualize concept-steering in input space, see Fig. 6, as well as to provide insights into the composite nature of the feature space, see Fig. 7.

## 2 Methods

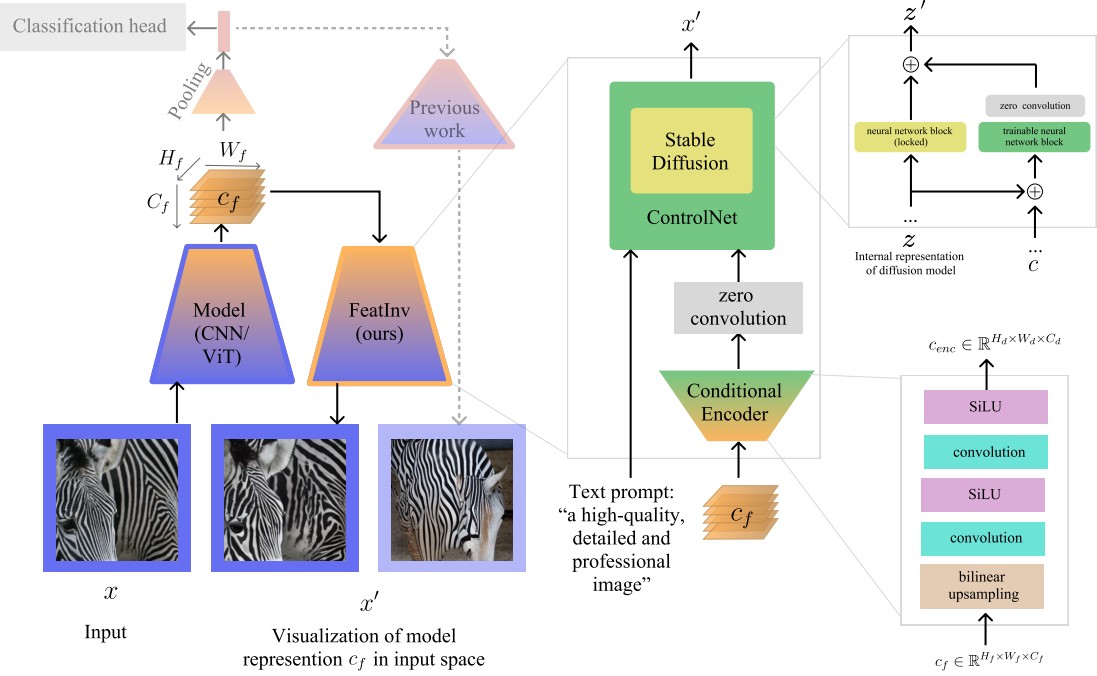

Figure 2: **Schematic overview of the *FeatInv* approach.** *Left:* Given a spatially resolved feature map $c_f$ of some pretrained model, we aim to infer an input $x'$ within the set of natural images, whose feature representation aligns as closely as possible with $c_f$, i.e., to learn a probabilistic mapping from feature space to input space. Previous work consider spatially pooled feature maps, whereas this work conditions on spatially resolved feature maps. *Middle:* We leverage a pretrained diffusion model, which gets conditioned on $c_f$ by means of a ControlNet architecture, which parametrizes an additive modification on top of the frozen diffusion model. *Right top:* The ControlNet adds trainable copies of blocks in the stable diffusion model, which are conditioned on the conditional input and added to the output of the original module, which is kept frozen. *Right bottom:* The feature map $c_f$ is processed through bilinear upsampling and a shallow convolutional encoder to serve as conditional input for the ControlNet.

**Approach** In this work, we propose a method called *FeatInv* to approximate an inverse mapping from a model's feature space to input space. Our method conditions a pretrained stable diffusion model on a spatially resolved feature map extracted from a pretrained CNN/ViT model of our choice. As described in detail in the next paragraph, the feature maps are provided as conditional information along with an unspecific text prompt ("a high-quality, detailed, and professional image") to a conditional diffusion model of the ControlNet (Zhang et al., 2023a) flavor. Importantly, rather than achieving a precise reconstruction of the original sample in input space, our goal is to infer high-fidelity, synthetic images whose feature representations align with those of the original image when passed through a pretrained CNN/ViT model.

**Architecture and training procedure** We use a ControlNet (Zhang et al., 2023a) architecture, building on a pretrained diffusion models, in our case a MiniSD (Pinkney, 2023) model operating at an input resolution of $256 \times 256$. The ControlNet is a popular approach to condition a pretrained diffusion model on dense inputs such as segmentation maps or depth maps. It leverages a pretrained (text-conditional) diffusion model, whose weights are kept frozen. The trainable part of the ControlNet model mimics the internal structure of the pretrained diffusion model, with additional layers introduced to incorporate conditioning inputs. These conditional inputs are processed by a dedicated encoder and inserted into the corresponding computational

blocks, where their outputs are added to those of the original diffusion model. Convolutional layers initialised to zero ensure that optimisation of the ControlNet model starts from the pretrained diffusion model.

**Conditional input encoder** An important design choice is the conditional input encoder, which maps the feature map (with shape $H_f \times W_f \times C_f$, where $H_f$,$W_f$,$C_f$ correspond to the height, width and channels of the feature map, respectively) to the diffusion model's internal representation space (with shape $H_d \times W_d \times C_d$). As a definite example for $224 \times 224$ input resolution, for the output of ResNet50's final convolutional block with $H_f = W_f = 7$, $C_f = 2048$ we learn a mapping to the diffusion model's internal representation space with $H_d = W_d = 32$ and $C_d = 320$. To this end, we first use bilinear upsampling to reach the target resolution. Then, we allow for a shallow CNN to learn a suitable mapping from the model's representation space to the diffusion model's representation space.

**Pooled vs. unpooled** To demonstrate superiority over prior work (Bordes et al., 2022), we also consider the case of pooled feature representations obtained from average-pooling spatial tokens/feature maps. In order to process them using the same pipeline as for conditioning on spatially resolved feature maps, we copy the $C_d$-dimensional input vector along $H_d$ and $W_d$ times to reach an input tensor with shape $H_d \times W_d \times C_d$ as before.

**Training** The ControlNet is trained using the same noise prediction objective as the original diffusion model (Ho et al., 2020). Control signals are injected at multiple layers throughout the network, rather than being restricted to the middle layers, allowing them to influence the denoising process at various stages. Training was conducted on the ImageNet training set with a batch size of 8 and a learning rate of 1e-5 using an AdamW optimizer with the stable diffusion model locked. The ControlNet was trained on ImageNet for three epochs over approximately 45 to 60 hours (depending on backbone) of compute time on two NVIDIA L40 GPUs. During the course of this project, about five times more models were trained until the described setup was reached.

**Full pipeline** We work with the original input resolution of the respective pretrained models, which varies between $224 \times 224$ and $384 \times 384$ for the considered models, see the Supplementary Material A.1 for a detailed breakdown. Even though the approach allows conditioning on any feature map, we restrict ourselves to the last spatially resolved feature map, i.e., directly before the pooling layer, and learn mappings to MiniSD's internal feature space. The MiniSD model always returns an image with resolution $256 \times 256$, which we upsample/downsample to the model's expect input resolution via bilinear upsampling/downsampling. The full generation pipeline is visualized in Fig. 2.

## 3 Related Work

**Conditional diffusion models** Achieving spatially controllable image generation while leveraging a pretrained diffusion model has been a very active area of research recently, see (Zhang et al., 2023b) for a recent review. Applications include the conditional generation of images from depth maps, normal maps or canny maps. Popular approaches in this direction include ControlNet (Zhang et al., 2023a) or GLIGEN (Li et al., 2023). The mapping from feature maps as conditional input is structurally similar to the mentioned cases of spatially controllable generation. However, there is a key distinction. In the previously mentioned cases, the conditional information typically matches the resolution of the input image. This often necessitates downsampling to reach the diffusion model's internal representation space. In contrast, commonly used classification models (including CNNs and vision transformers) leverage feature maps with a reduced spatial resolution. Consequently, the spatial resolution of the conditional information is typically lower dimensional than the diffusion model's internal representation space. This difference necessitates an upsampling operation before conditioning on feature maps.

**Feature visualization** The idea to reveal structures in feature space to understand what a neural network has learned is an old one. Approaches range from identifying input structures or samples that maximize the activation of certain feature neurons (Erhan et al., 2009; Nguyen et al., 2016) to approximate inversion of the mapping from input to features space (Zeiler, 2014). Our approach clearly stands in the tradition of the latter approach. Previous work has attempted to learn a deterministic mapping that "inverts" AlexNet feature maps (Dosovitskiy & Brox, 2016). This approach was recently extended to invert vision transformer representations

(Rathjens et al., 2024). In contrast, FeatInv learns a probabilistic mapping using state-of-the-art diffusion models and investigates state-of-the-art model architectures. Other approaches tackle the problem using invertible neural networks to connect VAE latent representations to input space (Rombach et al., 2020) and/or disentangle these representations using concept supervision (Esser et al., 2020). In contrast, FeatInv does not rely on a particular encoder/decoder structure but can use any pretrained neural network as encoder. The closest prior work to our approach is (Bordes et al., 2022), which also uses a diffusion model to learn a mapping from feature space to input space. However, it uses pooled representations as input, i.e. neglects the spatial resolution of the feature map. We argue that pooled representations are too coarse for many applications as they disregard the finegrained spatial structure of the feature space. Diffusion autoencoders (Preechakul et al., 2022) also explore how diffusion-based representations can be made both meaningful and decodable, but their latent codes are global vectors. In contrast, we condition directly on spatial feature maps to preserve fine-grained structure.

**Representation surgery** Finally, related feature inversion approaches have also been explored beyond computer vision, for example in natural language processing (Morris et al., 2023). Here, the ability to invert latent representations is seen as an essential component for representation surgery approaches (Avitan et al., 2025). *FeatInv* enables similar approaches for computer vision models.

## 4 Results

We investigate three models ResNet50 (He et al., 2016) (original torchvision weights), ConvNeXt (Liu et al., 2022b) and SwinV2 (Liu et al., 2022a) [1] all of which have been pretrained/finetuned on ImageNet1k. ConvNeXt and SwinV2 represent modern convolution-based and vision-transformer-based architectures, identified as strong backbones in (Goldblum et al., 2024). We include ResNet50 due to its widespread adoption. For each model, we train a conditional diffusion model conditioned on the representations of the last hidden layer before the final pooling layer to reconstruct the original input samples. Below, we report on quantitative and qualitative aspects of our findings.

### 4.1 Quantitative and qualitative comparison

**Experimental setup** For each ImageNet class, we reconstructed 10 validation set samples with FeatInv, resulting in 10,000 reconstructed samples. We adjust the diffusion model's control strength and guidance scale to optimize the match of classification outputs between original and reconstructed samples on the validation set, resulting in different control strengths and guidance scales for each model. Since our goal is to represent the feature space as accurately as possible, we also observed the cosine similarity between the feature maps of the original and reconstructed samples and observed a strong correlation between classification match and cosine similarity, which further supports our choice of parameters. In our experience, it is also possible to achieve good results with the same control strength and guidance scale for all three models, see the Supplementary Material A.2 for details. We reconstruct with a sample step size of 50. For each model this took roughly 12 hours on a single NVIDIA L40 GPU. We only generate one sample per feature map but it is also possible to generate multiple to observe the variability across reconstructions (given the same conditional input), see Supplementary Material A.3 for insights. The generated samples are assessed according to two complementary quality criteria, reconstruction quality and sample quality:

1. **Reconstruction quality** The encoded generated image should end up close to the feature representation of the original samples, which can be understood as a reconstruction objective that is implemented implicitly by conditioning the diffusion model on a chosen feature map. (1a) The most obvious metric is cosine similarity between both feature maps. However, not all parts of the feature space will be equally important for the downstream classifier. (1b) Most reliable measure is the classifier output itself. Focusing on top-predictions, one can also compare top-k predictions to the top prediction for the original sample. More general alignment measures between generated input and original feature representation are not helpful in this context, as we require a precise reconstruction of the original feature space for the downstream classifier above the layer under

---

[1] timm model weights: convnext_base.fb_in22k_ft_in1k, swinv2_base_window12to24_192to384_22kft1k.

consideration. A feature inversion method is deemed *faithful* if the reconstructed samples exhibit strong correspondence with the original samples, as assessed by the two metrics introduced above.

2. **Sample quality** We aim to generate samples within the set of high-fidelity natural images. In our case, this objective is implemented through the use of a pretrained diffusion model. Apart from qualitative assessments in the following sections, we rely on FID-scores as established measures to assess sample quality.

Table 1: **Reconstruction quality and image quality of the individual models**: For the three considered backbones, we indicate three performance metrics to assess the quality of our reconstructions: Cosine similarity in feature space (cosine-sim), calculated by averaging the cosine similarity of all superpixels while maintaining the dimensions of the original input (288 x 288) and upscaling the reconstruction (from 256 x 256), top5(1) matches using the top-1 prediction of the original sample as ground truth (top5(1) match), and FID scores (FID) to assess the quality of the generated samples. We consider generative models conditioned on unpooled feature maps (rows 1-3) and models conditioned on pooled feature maps (rows 4-6). The results indicate that the proposed approach produces faithful reconstructions with high fidelity that are both perceptually realistic and, when re-encoded, have similar values in the feature space as the original input.

| | Model | cosine-sim | top5(1) match | FID |
|---|---|---|---|---|
| unpooled | ResNet50 | 0.46 | 91% (70%) | 11.49 |
| | ConvNeXt | 0.61 | 94% (77%) | 8.20 |
| | SwinV2 | 0.53 | 95% (80%) | 12.69 |
| pooled | ResNet50 | 0.12 | 48% (23%) | 31.64 |
| | ConvNeXt | 0.19 | 44% (20%) | 31.67 |
| | SwinV2 | 0.16 | 47% (22%) | 49.04 |

**Reconstruction quality** Comparing identical models conditioned either on pooled or unpooled feature maps, not surprisingly unpooled models show a significantly higher reconstruction quality. Samples generated by models conditioned on unpooled feature maps show a very good alignment with the feature maps of the original samples (cosine similarities above 0.53 and top5 matching predictions of 94% or higher for the two modern vision backbones). Samples conditioned on pooled feature maps show some alignment but fail to accurately reconstruct the respective feature map and are therefore unreliable for investigations of structural properties of models. These findings support the hypothesis that the approach yield feature space reconstructions that closely match the original feature representations. We also directly compared our pooled results to the approach proposed by Bordes et al. (2022), see Supplementary Material A.5 for details.

**Sample quality** The corresponding class-dependent diffusion model achieves an FID score around 29, which is typically considered as good quality. The models conditioned on pooled representations still show acceptable FID scores between 31 and 49. Interestingly, models conditioned on unpooled representations show a significant increase in image quality with FID scores between 8 and 12. These results support the statement that the created samples were sampled from the space of high-fidelity natural images.

**Backbone comparison** Within each category (pooled vs. unpooled), there is a gap between the two most recent model architectures ConvNeXt and SwinV2, notwithstanding the architectural differences (CNN vs. Vision transformer) between the two, in comparison to the older Resnet50 models. The former achieve cosine similarities of .61 or higher and top5 matches of 94% or higher in the unpooled category. This suggests that there is a qualitative difference between the representations of ResNet50-representations and representations of more modern image backbones.

**Qualitative comparison** In Fig. 3, we present a qualitative comparison based on randomly selected samples. The visual impressions of ConvNeXt and SwinV2 reconstructions are similar to each other while also being close to the input sample despite the fact that they were trained on high-level semantic feature maps, i.e.,

without a reconstruction objective in input space. The ResNet50 reconstructions seem in many cases an interpretation of the sample's semantic content (see e.g. 2. toucan or 5. file), albeit with the correct spatial composition, while matching specific color composition and textures much less accurately than ConvNeXt and SwinV2. We primarily attribute the differences between ResNet and ConvNeXt/SwinV2 to the nature of the feature spaces themselves, stressing qualitative difference between modern architectures such as ConvNeXt and SwinV2 and older model architectures such as ResNet50, which are much more pronounced than the differences between different model architectures such as ViTs and CNNs. The samples obtained from conditioning on pooled feature representations often seem to capture overall semantic content of the image correctly (file, space shuttle, traffic light), but fail to reflect the details of the composition of the image. This can further be observed in the Supplementary Material A.4.

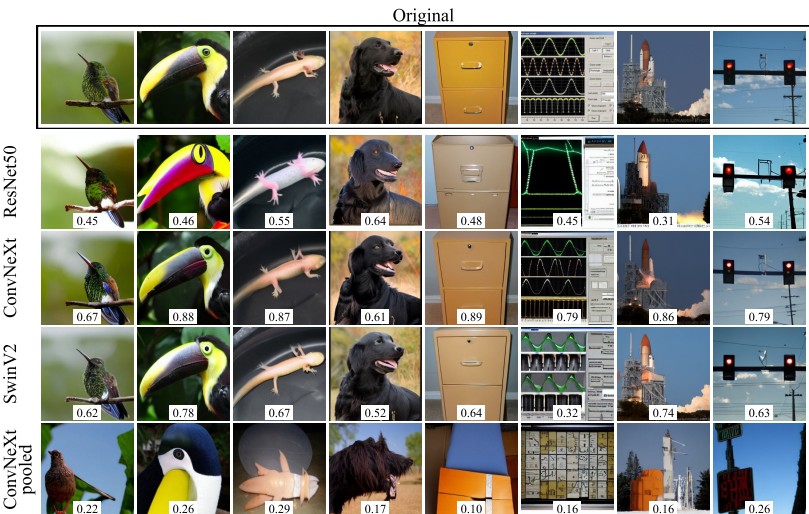

Figure 3: **Qualitative comparison of reconstructed samples** for the ResNet50, ConvNeXt, SwinV2 and ConvNeXt pooled models. The cosine similarity of the original feature map and that of the reconstruction is noted at the bottom edge of the images. The qualitative comparison confirms the insights from the quantitative analysis in Tab. 1. The two modern vision backbones, ConvNeXt and SwinV2, show reconstructions that resemble the original very closely, not only in terms of semantic content and spatial alignment but also in terms of color schemes and finegrained details. Semantic content and composition also mostly matches in case of the ResNet50, but not even the semantic content seems to be captured when using pooled representations (ConvNeXt pooled as an example).

**Conditioning on intermediate feature layers** The experiments so far focused on the last feature layer before the classifier in order to be able to visualize the features that ultimately lead to the classification decision (e.g., for *FeatInv-Viz* in Section 4.3). However, *FeatInv* can also applied to intermediate feature layers. For demonstration, we trained three additional *FeatInv* models conditioned on the respective last feature of the first, second, and third stage within the ConvNeXt model. As expected, conditioning on earlier feature provides more specific spatial context and therefore leads to more accurate and detailed reconstructions, see the qualitative examples in Figure 14 in the supplemental material. As in Table 1, we can also assess the reconstruction quality and the image quality quantitatively. The main difference compared to Table 1 is the ability to consider not only the cosine similarity compared to the feature layer but also compared to other feature layers, which were here taken identical to the three intermediate layers used for conditioning above. The results of this experiment are compiled in Figure 4. Most notably, the faithfulness extends beyond the layer that was used for conditioning, i.e., the models generally achieve good reconstruction across all considered feature layers. In line with expectations, the similarity increases when conditioning on earlier feature layers as they provide more finegrained information on the composition of the image. Furthermore, it is interesting to note that the similarity increases consistently for all models with the later stages the feature maps were extracted from (except the last one). This suggests that the last feature maps from the second and

third stages are less sensitive to reconstruction-related pixel deviations and instead represent more robust, semantically stable image features.

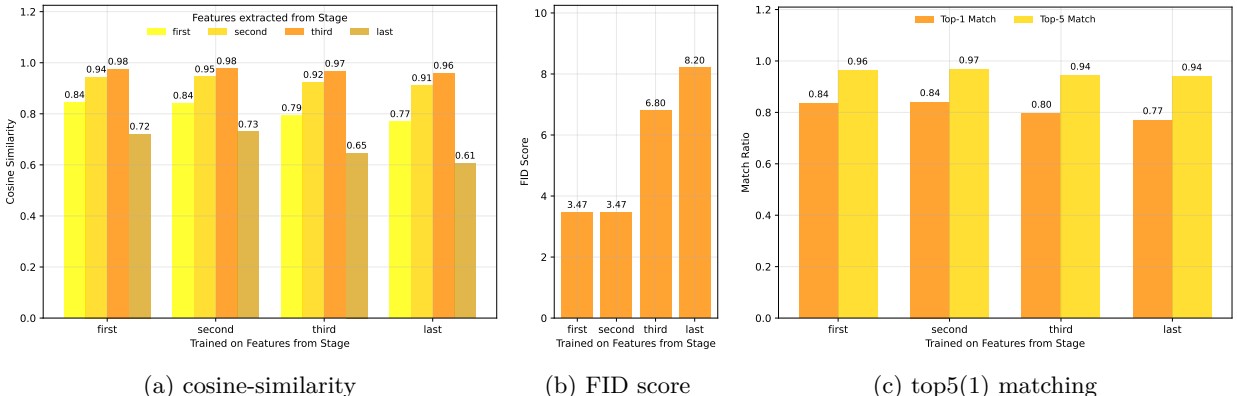

(a) cosine-similarity     (b) FID score     (c) top5(1) matching

Figure 4: **Comparison of the cosine similarity, FID score and top5(1)match for the models trained on the last feature maps of the first, second, and third stage of ConvNeXt** calculated by averaging the cosine similarity of all superpixels. The models trained on earlier features and reconstructing them perform better in terms of cosine-sim, FID score and matchings than the one which was trained on the very last feature map

## 4.2 Robustness evaluation

**Cross-model evaluation** To assess the robustness of the presented results, we carry out cross-model comparisons where we measure model performance based on samples generated by conditioning on the feature map extract from a different model. The results for this experiment are compiled in Tab. 2. It turns out that all three sets of samples (conditioned on features generated by the three different backbones) transfer quite remarkably to other models.

Table 2: **Cross-model evaluation**: Percentage of matching of the actual predictions (top5/top1) and the predictions based on the reconstructions for different models. The FeatInv models based on the ResNet50, ConvNeXt and SwinV2 features were used for the reconstruction and evaluated by the same three models.

| conditioned on | evaluated by | | |
|---|---|---|---|
| | **ResNet50** | **ConvNeXt** | **SwinV2** |
| **ResNet50** | 91% / 70% | 90% / 68% | 91% / 71% |
| **ConvNeXt** | 92% / 73% | 94% / 77% | 96% / 81% |
| **SwinV2** | 94% / 77% | 94% / 78% | 95% / 80% |

**Evaluation with OOD samples** In Fig. 5, we qualitatively test the robustness of our findings by conditioning on samples that are slightly out of the model scope of the original models finetuned on ImageNet. To this end, we use the BAID (Yi et al., 2023) dataset, which differs in style from the samples in ImageNet. We reconstructed 10,000 images from this dataset. The ControlNet trained on ConvNeXt features still shows a good reconstruction quality of the semantic content but the style of the reconstruction and the original image differ more strongly than in the in-distribution case in Fig. 3. The average cosine similarity is 0.48 and the FID value is 12.98. Nevertheless, the results speak for the robustness of the proposed approach.

**Evaluation with adversarial images** We tested FeatInv with adversarial images, which are often misclassified by models trained on ImageNet. For this, we used the ImageNet-A (Hendrycks et al. (2019)) dataset, which contains 7,500 samples. In the case of misclassification, the model can be misled by image characteristics such as small objects or confusing patterns. We reconstructed all images in this dataset using

Original

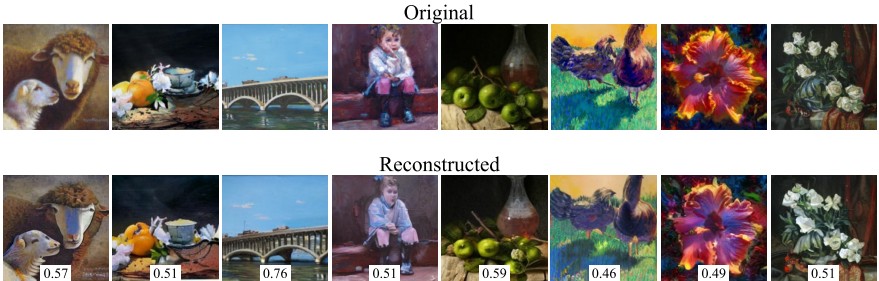

Reconstructed

Figure 5: **Reconstruction of OOD samples**. Comparison of randomly selected original and generated samples from the BAID (Yi et al., 2023) dataset, which differs in style from the samples in ImageNet. The basis for the reconstruction was the ControlNet trained on ConvNeXt features, which received the ConvNeXt features of the samples shown as input. The cosine similarity between the original feature map and that of the reconstruction is noted at the bottom edge of the images

our FeatInv model trained on ImageNet ConNeXt feature maps. The average cosine similarity is 0.56 and the FID score is 12.44. FeatInv is therefore able to reconstruct these unknown difficult images to a good degree. The top5(1) classification match is 0.71(0.38), which means that more than a third of the images are classified as the original image, suggesting that parts of the image relevant for classification are reconstructed in a meaningful way. Nevertheless, these matching values are smaller than those of the reconstructed simple images, indicating that the adversarial images are more complex and difficult to reconstruct.

**Evaluation with corrupted images** We also tested FeatInv with corrupted images. To this end, we used the ImageNet-C dataset (Hendrycks & Dietterich (2019)), which contains various categories of image corruption of varying degrees of intensity. Here, we focused on the weakest (level 1) and the strongest (level 5) corruption levels. In the supplementary material, we present the FID scores for the respective categories and the respective cosine similarity values as well as both the top 5(1) accuracy matches, which correspond to the cosine similarity results, and qualitative results (see Supplementary Material C). As expected, all corruptions have a negative effect on both reconstruction quality and sample quality. Noise and contrast in particular have a high negative impact on the scores and cannot be reproduced well by FeatInv. However, it is worth noting that level 5 corruption represent represent a significant distortion of the original images and therefore represents a challenging robustness scenario.

### 4.3   Application: *FeatInv-Viz* – Visualizing concept steering in input space

**Concept steering in input space** In generative NLP, steering is sometimes used to verify concept interpretations by reducing or magnifying concepts in the model activations and observing how this changes the generated output text, as famously demonstrated at the example of the Golden Gate concept (Bricken et al., 2024) in Claude 3 Sonnet. This approach is not directly applicable to vision classifiers. However, with our method of inverting model representations from feature to input space, we can observe the effect of concept steering within hidden model activations in the input representation space instead of the output. This enables a novel method for concept visualization, with benefits over existing approaches (see below).

**Concept definition** Concepts are typically defined as structures in feature space such as individual neurons, single directions or multi-dimensional subspaces. Many concept-based XAI methods define a way to decompose a feature vector into concepts from a dictionary/concept bank (Fel et al., 2023). In this work, we use concepts from multi-dimensional concept discovery (MCD) (Vielhaben et al., 2023), which defines concepts as linear subspaces in feature space. Nevertheless, our approach is applicable to any concept discovery method.

**Concept visualization through attenuated feature maps** A common challenge for unsupervised concept discovery methods is inferring the meaning of discovered concepts. To address this, we steer a concept in feature space and observe the effect in input space. Specifically, we attenuate coefficients for the concept under consideration to 25%, see the Supplementary Material D for details. Then, we use FeatInv to map the original and the modified feature map to input space using identical random seeds for the diffusion process.

By comparing the resulting images, we gain insights into how the concept is expressed in input space. We call this method *FeatInv-Viz* and present it in Algorithm 1.

---

**Algorithm 1:** *FeatInv-Viz*: Visualization of concept steering in input space

---

**Input:** Model $m$, concept decomposition $\phi = \sum_i \phi_i$, concept with id $c$
**Output:** Visualization of concept $c$ in input space
**Notation:** $x \in \mathbb{R}^{3 \times H \times W}$ where $x^{(j)}$ refers to color channels with $j \in \{R, G, B\}$
$\phi' \leftarrow \sum_{i \neq c} \phi_i + 0.25 \cdot \phi_c$ ;                                    // Attenuated feature map
**for** $i = 1$ **to** $n$ **do**
    $s_i \leftarrow \text{RandomSeed}()$
    $x_i \leftarrow \text{FeatInv}(\phi, seed = s_i)$ ;                                    // Original reconstruction
    $x_i' \leftarrow \text{FeatInv}(\phi', seed = s_i)$ ;                                    // Attenuated reconstruction
    $\Delta_i \leftarrow \sqrt{\sum_{j \in \{R,G,B\}} (x_i^{(j)} - x_i'^{(j)})^2}$ ;                                    // Euclidean distance
**return** $median\{\Delta_i\}_{i=1}^n$ ;                                    // Median along sample axis

---

**Exemplary results** Fig. 6 shows exemplary concept steering visualizations for four samples from the Indigo Bunting class. Here, we decomposed ConvNeXt's feature space into three linear concept subspaces. *FeatInv-Viz* provides a visualization of these concepts in input space. The method provides a very finegrained visualization of which specific regions in input space change upon steering each concept in feature space. More examples can be found in the Supplementary Material D.1.

**Benefits** We emphasize that *FeatInv-Viz* extends commonly used concept activation maps in two ways: First, it provides a finegrained visualization rather than a coarse upscaling Bau et al. (2017); Vielhaben et al. (2023) of a lower-resolution feature map. Second, it goes beyond merely verifying alignment with a predefined concepts Bau et al. (2017), by providing counterfactual information from targeted feature-map manipulations.

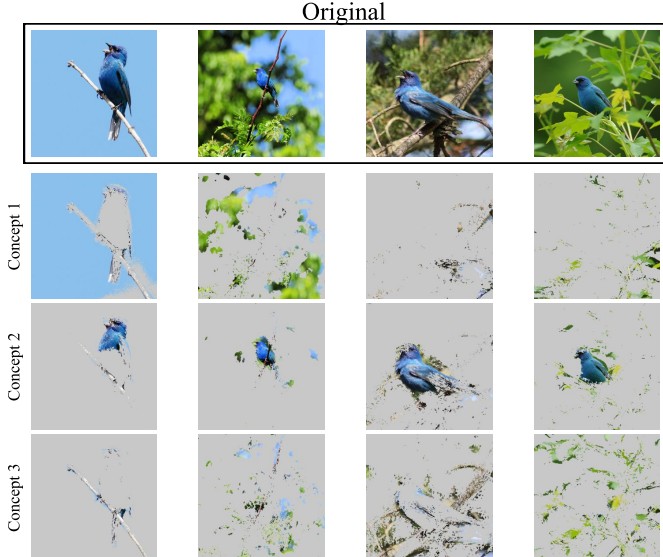

Figure 6: ***FeatInv-Viz* visualization of three concepts** identified within ConvNeXt's feature space of the Indigo Bunting class, which can be associated with sky/background, bird head/breast and branches/leaves. For the visualization we normalize the respective outputs of Algorithm 1 and threshold it below 0.33 as a binary mask to indicate unaffected regions of the image.

## 4.4 Application: Investigating the composite nature of the feature space

In NLP, well-known examples of feature-space arithmetic – e.g. king − man + woman = queen Mikolov et al. (2013) – have shaped our understanding of embedding geometries. FeatInv offers insights into the composite nature of the feature space in vision models by conditioning on feature maps from two samples. In particular, we investigate the effect of convex linear superpositions of two feature maps. To this end we linearly interpolate between the feature representations of two input samples and visualize reconstructions for different weighted combinations, as shown in Fig. 7. We also indicate the cosine similarity between the reconstruction and the weighted feature map, which is highest for the original feature maps and typically reaches its lowest value for the equally weighted interpolated feature map. This can be seen as an indication that the weighted average of two feature maps is in general not a well-defined operation. Nevertheless, foreground objects from one image and background from a second, seem to be reasonably combined through linear superposition (see e.g. bird, landscape).

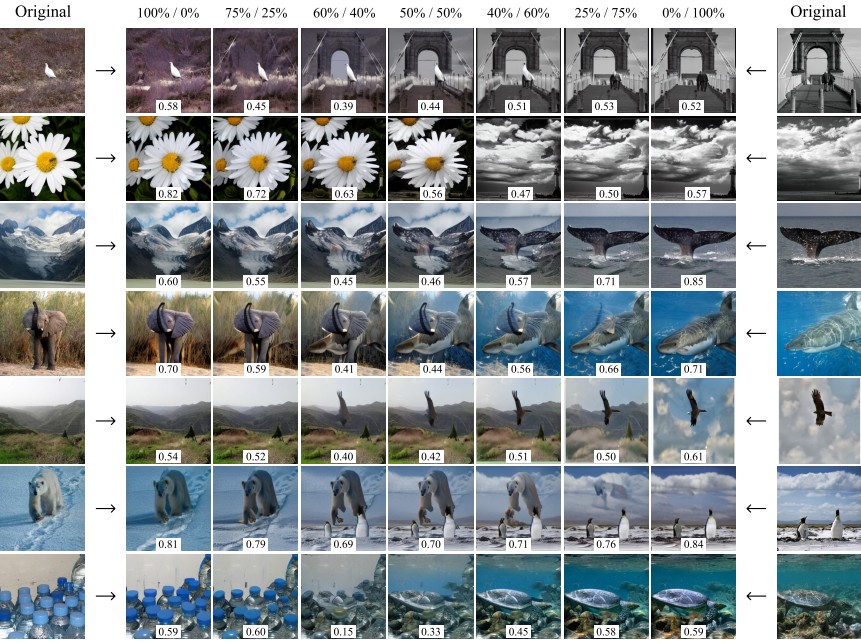

Figure 7: **Reconstructions from weighted combinations of two ConvNeXt feature maps**. The cosine similarity between the weighted feature map and that of the reconstruction is noted at the bottom edge of the images.

In Fig. 8, we show spatially composed combinations of two feature maps. The results indicate that feature maps exhibit a very local influence, which aligns well with the simple upscaling of the feature map resolution to the input resolution.

## 4.5 Limitations and future work

Our work is subject to different limitations, which provide directions for future investigations: First, the present work focuses exclusively on the domain of natural images. It would be very instructive to extend the approach to other domains, such as medical imaging. Second, the proposed approach building on the ControlNet method, builds on a pretrained diffusion model, which might not be readily available in any application contexts. Third, every model and layer choice requires training a dedicated FeatInv model, which represents a computation hurdle. First experiments and the results in Tab. 2 indicate that finetuning could be beneficial to alleviate this issue. Finally, both application scenarios rely on modifications of the feature space. In order to obtain reliable results, it would be instrumental to introduce measures to detect input samples, i.e., feature maps that are outside the scope of the model.

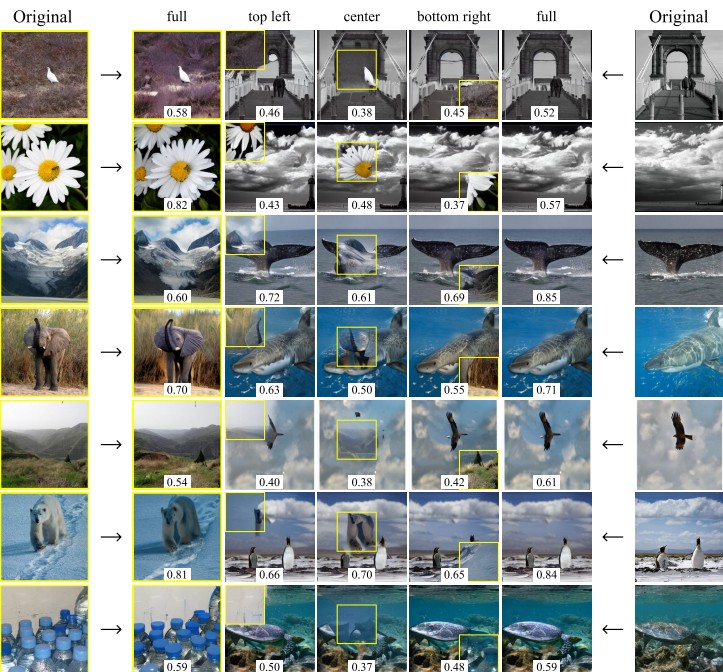

Figure 8: **Reconstructions of spatially composed mixtures of two ConvNeXt feature maps**. The cosine similarity between the manipulated map and that of the reconstruction is noted at the bottom edge of the images. The yellow outlines show the part of the feature map that was manipulated

## 5 Summary and Discussion

In this work, we address the problem of obtaining insights into the structure of a given model's feature map by means of a learned probabilistic mapping from feature space to input space, implemented as a conditional diffusion model. We demonstrate the feasibility of training such a model in a ControlNet-style achieving very accurate and robust reconstruction results across different model architectures. We present two possible applications both of which relate to gaining inside into manipulated feature maps. However, we believe that the proposed approach could be widely applicable to further applications. We envision a potentially positive societal impact through improved model understanding, along the lines of the concept steering use case. The source code underlying our investigations is available at `https://github.com/AI4HealthUOL/FeatInv`.

## Acknowledgements

This work was supported by the Bundesministerium für Forschung, Technologie und Raumfahrt (BMFTR) under Project INGVER, grant number 01KX2419.

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

# A Ablation studies

## A.1 Additional models

For the ablation studies, we consider ResNet50B[2] (Wightman et al., 2021) as second ResNet-model to study the impact of the training procedure, in addition to the models studied in the main text.

**Choice of conditional encoder** Next to the bilinear upsampling followed by a shallow CNN, we explore a different design choice for the conditional encoder: To this end, we explore the use of cross-attention as used in the Perceiver architecture (Jaegle et al., 2021), which uses a learnable representation of predefined size and connects it via cross-attention to the representation that serves as input for the mapping. Unlike the previous approach, this approach is not subject to any locality assumptions and therefore the most flexible approach that is in particular suitable for model architectures without built-in locality assumptions such as vision transformers. We carry out a comparison between the two conditional encoder models for the case of the SwinV2 ViT, where we expect the impact to be most pronounced as the ViT operates on visual tokens, which might not align well with the upscaling operation in the convolutional encoder. On the contrary, our results indicates that the convolutional encoder yields better classification results and therefore for the following experiments we restrict ourselves to the convolutional encoder.

**Class-conditional baseline** We tested the models on an unconditional baseline to see if MiniSD is able to generate good representations of the classes and if they can be correctly classified. To define the classes for the input prompt as precisely as possible, we use the WordNet hierarchy and create the prompt as follows: 'a high-quality, detailed, and professional image of a CLASS, which is a kind of SUPERCLASS' as for example 'a high-quality, detailed, and professional image of a tench, which is a kind of cyprinid'. The unconditional baseline created consists of 10 samples per class, i.e. 10.000 samples in total.

Table 3: **Additional model insights**: For the different backbones, we indicate the corresponding canonical input sizes as well as the top5(1) accuracy on an unconditional dataset, created by MiniSD. Since only the base model is used for the prediction of these unconditional samples, the results do not differ between unpooled and pooled as well as for the different SwinV2 architectures. We show three performance metrics: Cosine similarity in feature space (cosine-sim), calculated by taking the mean of the cosine similarity of all superpixels, top5(1) matches using the top1 prediction of the original sample as ground truth (top5(1) match) and FID-scores (FID) to assess the quality of the generated samples. We consider generative models conditioned on unpooled feature maps (rows 1-5) and models conditioned on pooled feature maps (rows 6-9). While reconstructions were generated using model-specific values for control strength and guidance scale, we also observed that good results can be achieved with uniform values. A control strength of 1.5(1) and a guidance scale of 8(1.75) for (un)pooled conditional input seemed to work well for this scenario.

| | Model | input | MiniSD top5(1) | cosine-sim | top5(1) match | FID |
|---|---|---|---|---|---|---|
| unpooled | ResNet50 | $224 \times 224$ | 89% (68%) | 0.46 | 91% (70%) | 11.49 |
| | ResNet50B | | 88% (65%) | 0.50 | 90% (70%) | 11.51 |
| | ConvNeXt | $288 \times 288$ | 92% (71%) | 0.61 | 94% (77%) | 8.20 |
| | SwinV2 | $384 \times 384$ | 93% (72%) | 0.53 | 95% (80%) | 12.69 |
| | SwinV2(Perceiver) | | | 0.36 | 76% (53%) | 22.65 |
| pooled | ResNet50 | $224 \times 224$ | 89% (68%) | 0.12 | 48% (23%) | 31.64 |
| | ResNet50B | | 88% (65%) | 0.14 | 44% (21%) | 37.44 |
| | ConvNeXt | $288 \times 288$ | 92% (71%) | 0.19 | 44% (20%) | 31.67 |
| | SwinV2 | $384 \times 384$ | 93% (72%) | 0.16 | 47% (22%) | 49.04 |

---

[2]timm model weights: resnet50.a1_in1k

## A.2 Control Strength vs. Guidance Scale

In Fig. 9, we present several original images alongside their respective reconstructions, generated using the model trained with the ConvNeXt backbone. These samples were produced with varying levels of control strength and guidance scale. Since both significantly influences the output, we evaluated the model's performance across different scale settings, again using the ConvNeXt-based model. To quantify the quality of the generated images, we calculated the Top-1 classification match as well as the cosine similarity, illustrated in Fig. 10 and 11. Both correlate strongly and show the same goog performing ranges. This suggests that the unpooled feature map requires minimal text guidance, whereas for the pooled variant, guidance is of substantial importance, as natural looking images cannot be generated without its influence.

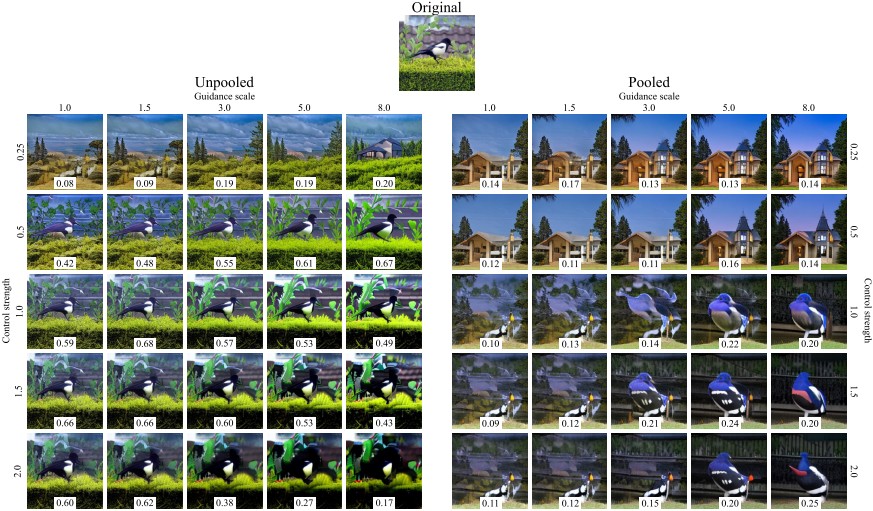

(a) Reconstructions for class Magpie

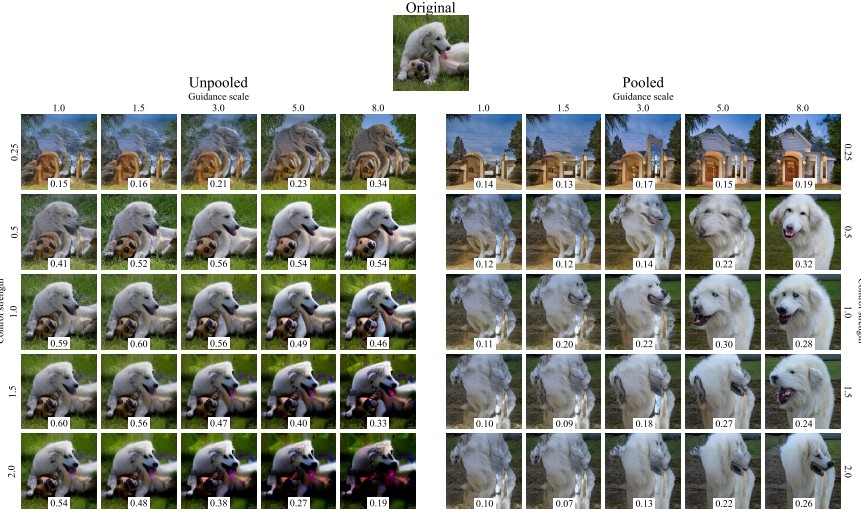

(b) Reconstructions for class Kuvasz

Figure 9: **Unpooled and pooled reconstructions with different control strength and guidance scales from ConvNeXt feature maps for class magpie and kuvasz**. The cosine similarity between the original feature map and that of the reconstruction is noted at the bottom edge of the images. A low control strength already results in a sufficient reconstruction. While increasing the guidance scale in unpooled samples results in higher color saturation and more distorted object shapes, the same increase improves object fidelity in pooled feature maps. This trend is also roughly reflected in the corresponding cosine similarity.

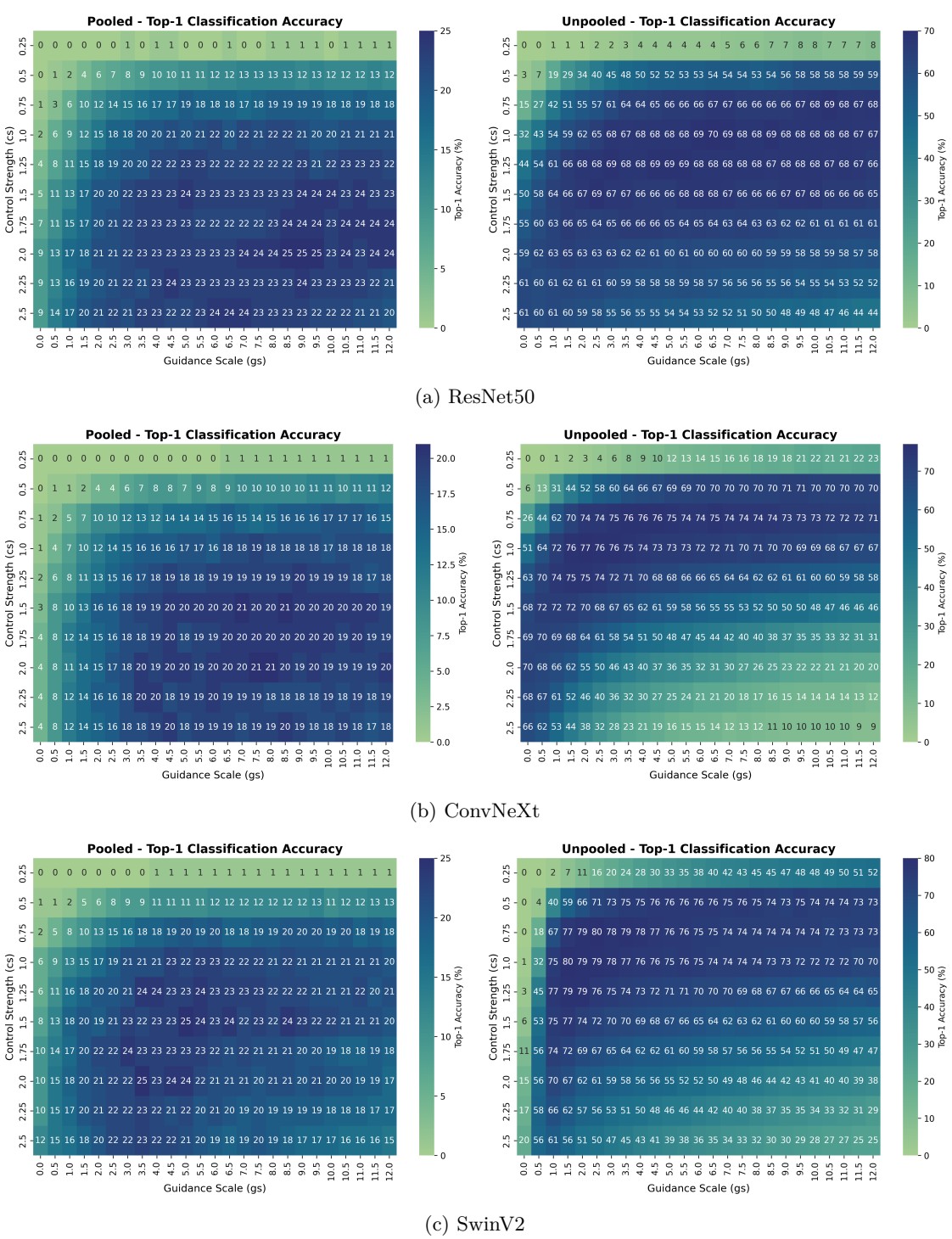

Figure 10: **Impact of control strength and guidance scale on generation quality (top-1 accuracy).** Top-1 classification accuracy for ResNet50, ConvNeXt and SwinV2 backbone with and without pooling. One image per class was reconstructed for each combination. Higher values indicate better performance. The unpooled model does not seem to rely on guidance as much as the pooled model.

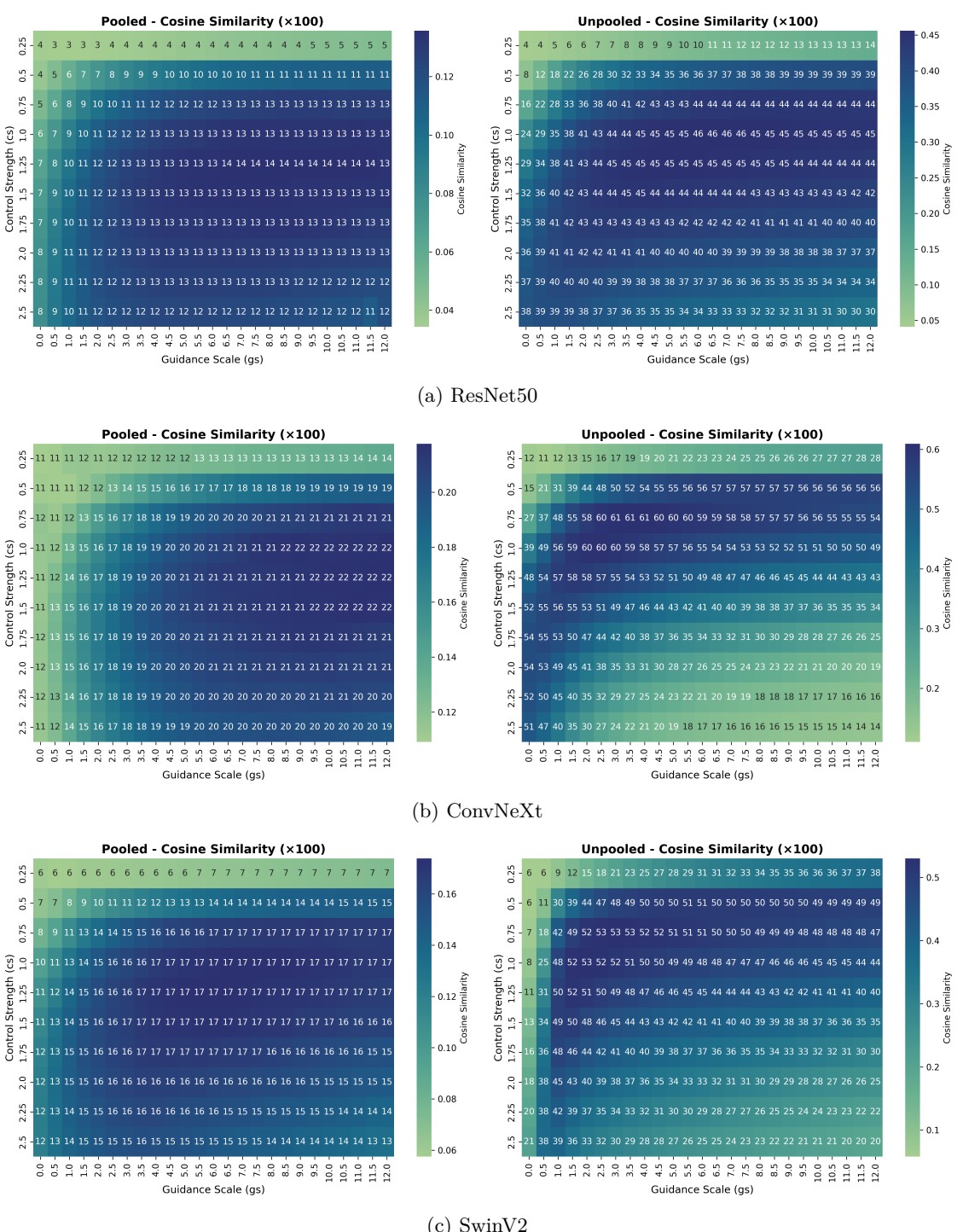

(a) ResNet50

(b) ConvNeXt

(c) SwinV2

Figure 11: **Impact of control strength and guidance scale on generation quality (cosine-sim).** Cosine similarity (x100) for ResNet50, ConvNeXt and SwinV2 backbone with and without pooling. One image per class was reconstructed for each combination. Higher values indicate better performance. The unpooled model does not seem to rely on guidance as much as the pooled model.

## A.3 Variability across reconstructions

As the model we use for reconstruction is a probabilistic model, we show the variance of the generations qualitatively when different seeds are used. This can be seen in Figure 12. The observed variability supports

the necessity of a probabilistic rather than a deterministic mapping for the ill-posed problem of mapping from feature space to input space.

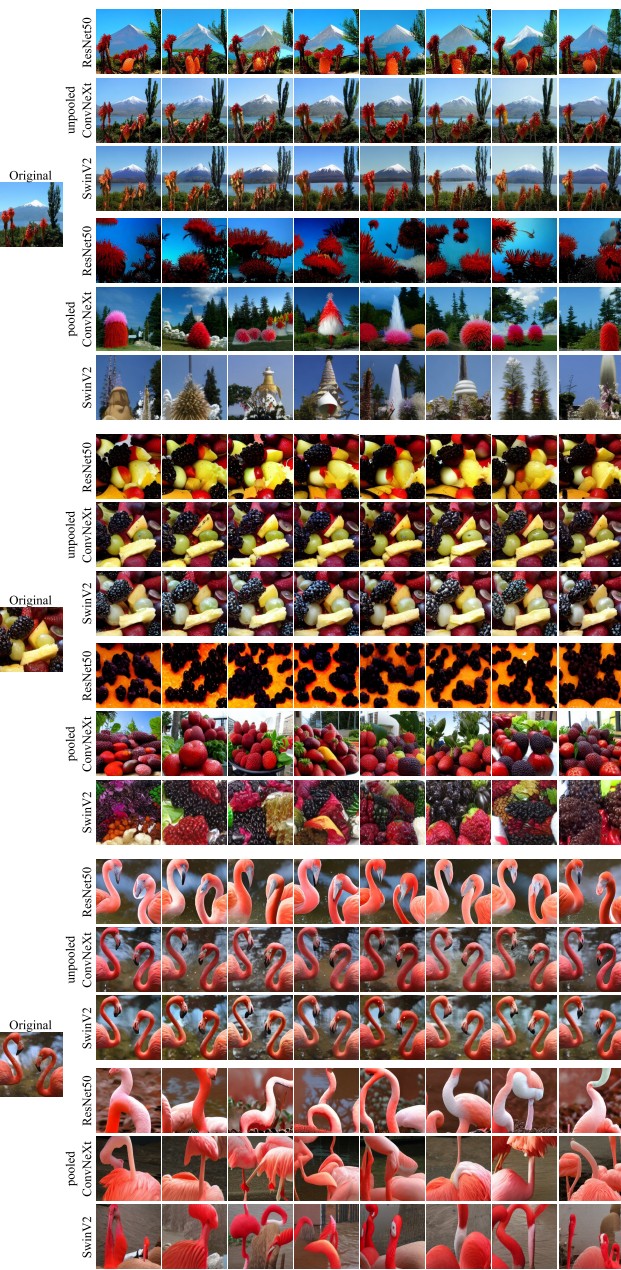

Figure 12: **Reconstruction variations** of feature maps from ResNet50, ConvNeXt, and SwinV2, each unpooled and pooled for the classes volcano, strawberry, and flamingo. It is noticeable that the unpooled reconstructions mostly resemble each other; only ResNet50 displays noticeable differences. However, the reconstructions on pooled feature maps are different. Here, the images appear very diverse, even though they are semantically similar. For example, the volcano class in ResNet50 pooled shows a type of underwater landscape consistently, while the pooled images in ConvNext all show red bushes. Consistency between the reconstructions of the pooled feature maps can also be seen in the other classes, although these are never as strong as those of the unpooled feature maps.

## A.4 Generation on unpooled vs. pooled feature maps

We analyzed the generated samples for each model using both unpooled and pooled feature maps. As shown in Tables 1 and 3, the samples generated from pooled feature maps consistently exhibit lower visual quality compared to those generated from unpooled feature maps. This indicates that preserving spatial detail during feature extraction is crucial for high-quality image generation. Fig. 13 illustrates this difference with examples from the class zebra, clearly highlighting the superior fidelity of samples generated from unpooled features.

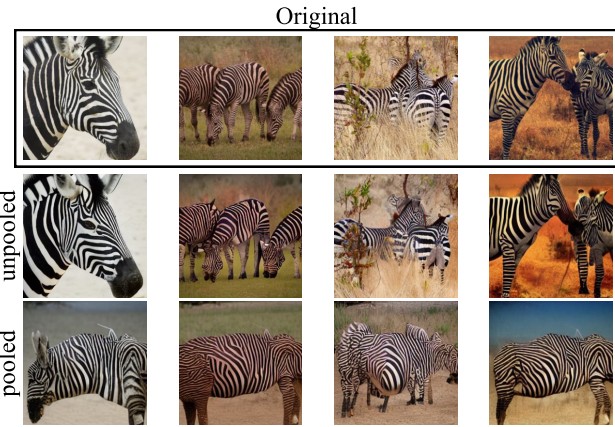

Figure 13: **Comparison of unpooled and pooled generated samples** showing examples of the class Zebra using. All samples were generated using the model trained with the ConvNeXt backbone. The unpooled feature maps result in sharper and more realistic generations, while the pooled versions show a noticeable loss in detail and structure, mostly showing a general context of the original image. The cosine similarity between the original feature map and that of the reconstruction is noted at the bottom edge of the images

## A.5 Comparison to RCDM

**Approach** In this section, we provide a direct comparison to the RCDM approach Bordes et al. (2022) as our closest competitor. The RCDM approach conditions on pooled feature representations and can therefore be most meaningfully compared to *FeatInv* conditioned on pooled feature maps, which receives the same input. We stress at this point, that this experiment facilitates a comparison of different approaches to train diffusion models conditioned on pooled feature maps but does not touch the central novelty of the submission, i.e., conditioning on unpooled feature maps. Whereas RCDM trains a conditional diffusion model from scratch, *FeatInv*'s ControlNet approach leverages a pretrained diffusion model.

**ResNet50** We present the results for a RCDM model Bordes et al. (2022) conditioned on pooled ResNet50 feature maps, which produces samples at a resolution of 128×128 pixels. Here, we leverage model weights provided by the original authors. To evaluate the quality of the generated images, we computed the same metrics as for the *FeatInv* models presented in the main text, as shown in Table 4. The results are slightly non-trivial to compare between RCDM and *FeatInv* as the output resolutions differ (128×128 vs. 256×256), which leaves the option of either evaluating RCDM outputs at their native resolution or scaling it up to 224×224. Irrespective of the precise evaluation procedure, RCDM shows slightly higher cosine similarities (0.16-0.29 vs. 0.12) and higher FID scores (13.0-28.3 vs. 31.6) than pooled *FeatInv* models. However, these values remain lower than those achieved by our approach using unpooled feature maps.

**ConvNeXt** We also trained an RCDM model ourselves based on ConvNeXt as ResNet50 showed the worst results across all investigated models in the main text. We therefore used the official RCDM code repository to train a RCDM model conditioned on pooled ConvNeXt feature maps, as before on a resolution 128×128. After 95 hours of training on four L40 GPUs with a batch size of eight, the model had reached 520,000 steps. It performs worse than our *FeatInv* equivalents which shows that it either did not fully convergence, indicating that further training would be needed or is not capable of recreating matching samples based on

the ConvNeXt features. Our *FeatInv* models, which were trained using a smaller batch size of two for less time, saw less images after 1,920,000 steps and performed better. This demonstrates the benefit of using a pre-trained diffusion model rather than training one from scratch.

**Conclusion** While RCDM is technically capable of scaling to higher resolutions than 128x128, doing so requires long training times. In our experiments, we trained RCDM at half the resolution of our *FeatInv* models and for a longer duration, yet its performance still lagged behind *FeatInv*. In contrast, *FeatInv* could be adapted to even higher input resolutions as it leverages a pretrained diffusion model. Results are difficult to comparable directly due to different input resolutions. RCDM seems slightly superior in the case of ResNet50, but clearly inferior in the case of ConvNeXt. In any case, the RCDM approach is not able to deliver models conditioned on unpooled feature maps, which yield clearly superior reconstruction quality and better FID scores. This clearly positions *FeatInv* as the more versatile approach.

Table 4: **Comparison between *FeatInv* and RCDM**: We computed cosine similarity between feature maps of generated and original images. Since there is a significant gap between the dimensions of the original and generated samples, we also examined the results when both were scaled to the same low input size (resulting in 4x4 feature maps), which increased the similarity. Classification accuracy and FID score likewise improved when both inputs were resized to the lower resolution. The ConvNeXt RCDM does not reach the results of our *FeatInv* equivalent after seeing more than the same amount of images in training.

|  | Model | input | cosine-sim | top5(1) match | FID |
|---|---|---|---|---|---|
| RCDM | ResNet50 | $128 \times 128$ | 0.29 | 69% (42%) | 13.02 |
| | ResNet50 | $224 \times 224$ | 0.16 | 65% (37%) | 28.26 |
| | ConvNeXt | $128 \times 128$ | 0.12 | 26% (10%) | 37.20 |
| | ConvNeXt | $224 \times 224$ | 0.14 | 26% (10%) | 57.46 |
| FeatInv | ResNet50 | $224 \times 224$ | 0.46 | 91% (70%) | 11.49 |
| | ConvNeXt | $288 \times 288$ | 0.61 | 94% (77%) | 8.20 |
| | ResNet50 pooled | $224 \times 224$ | 0.12 | 48% (23%) | 31.64 |
| | ConvNeXt pooled | $288 \times 288$ | 0.19 | 44% (20%) | 31.67 |

# B    Conditioning on intermediate feature layers

To condition on intermediate feature layers, we adjusted the input and hidden dimensions of our input encoder to the different dimensions in the model. Unlike the model on the last feature map, each of these models was trained for only one epoch. We observed that training on the last feature maps from early ConvNeXt stages in particular led to good reconstruction results much faster than training on features from the last layer. For these models, we also reconstructed 10 validation images per class (see Figure 14) and calculated cosine similarity, FID and matching accuracies (see Figure 4). For the reconstruction we provide the last feature map from the stage the model was trained on.

In Figure 14, it can be seen that the models that reconstruct earlier feature maps work more accurately and display details better (e.g., watermark text). Colors and edges within the image are also displayed better.

# C    Reconstruction of corrupted images

We used FeatInv to reconstruct corrupted images as described in Section 4.1. Both the FID score (see Figure 15) and cosine similarity (see Figure 16) suggest that FeatInv can only represent these manipulation up to a certain level. We show the top5(1) accuracy matches for the corrupted images and their reconstructions (see Figure 17(18)). In addition we present some examples of what the reconstruction of the different categories with FeatInv looks like (see Figures 19, 20, 21, 22 and 23).

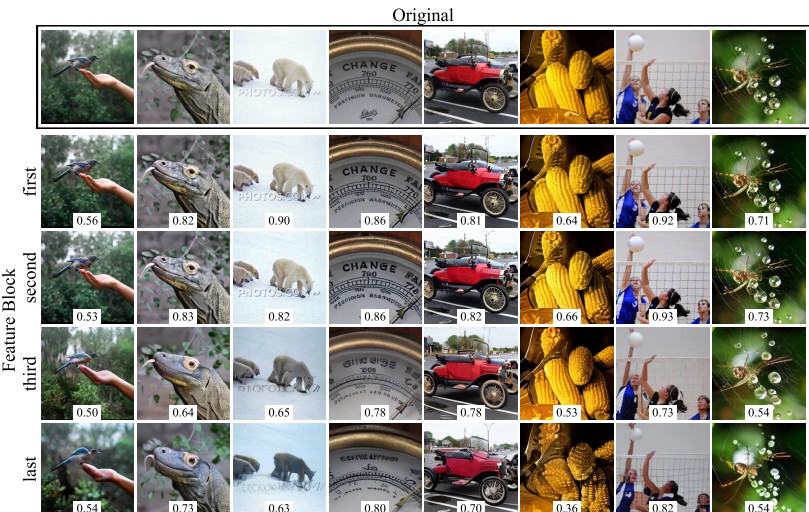

Figure 14: **Comparison of reconstructions of models trained on the last feature maps from different ConvNeXt stages** with dimensions (128, 72, 72), (256, 36, 36), and (512, 18, 18). The cosine similarity of the original feature map and that of the reconstruction is noted at the bottom edge of the images.

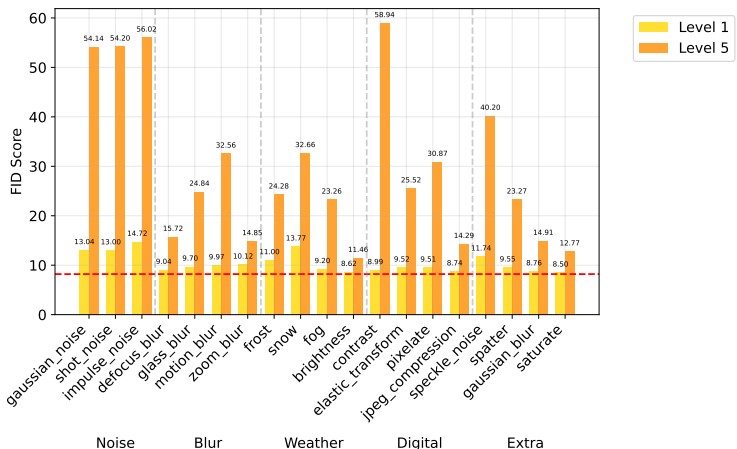

Figure 15: **Comparison of the FID scores of the ImageNet-C categories.** While a slight corruption of the images, as in level 1, only leads to a slight increase in the FID score, a high corruption has a significant impact on the score. The higher the corruption, the more difficult it is for FeatInv to reconstruct the corrupted images. It should be noted that the model was not trained on any of these corruptions. Noise and contrast in particular have a high negative impact on the scores and cannot be reproduced well by FeatInv.

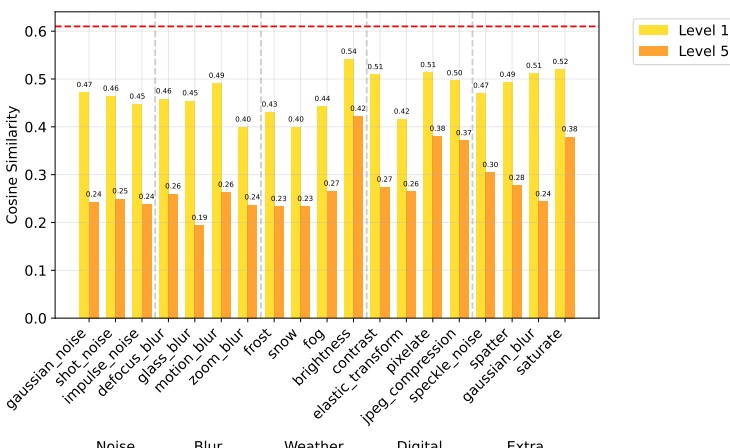

Figure 16: **Comparison of the cosine similarity of the feature maps of the corrupted input images and their reconstructed images.** Both minor (level 1) and major corruption (level 5) have a strong influence on the cosine similarity of the final feature maps and cause them to drop. It should be noted again that the model was not trained on any of these corruptions. Unlike the FID scores of the images in these different categories, there does not seem to be any category in which the change differs significantly from that of the other categories.

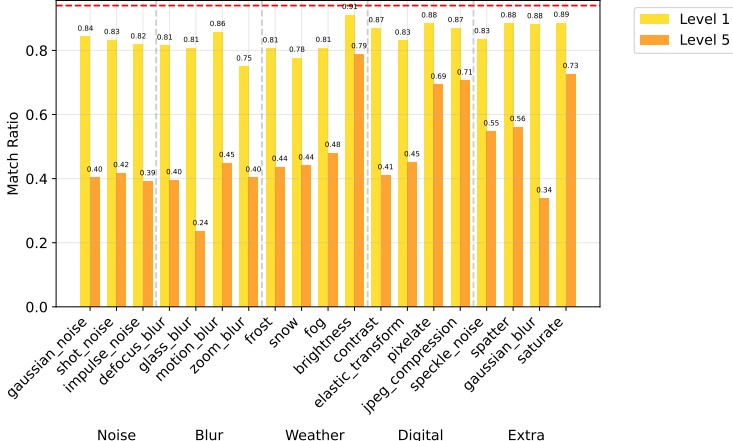

Figure 17: **Comparison of the top5 accuracy matches of the ImageNet-C categories.** While a slight corruption such as in level 1 achieves almost the same value in top5 accuracy matching as the unaltered images, a stronger corruption such as in level 5 has a much greater negative impact on the matching. Compared to the cosine similarity in Figure 16, the matching works unexpectedly well, which is most likely due to the fact that the model assigns the classes of the two images, one of which was reconstructed by featinv and may not contain the complete corruption, to the same class. The result is different when there is a large difference in the reconstruction.

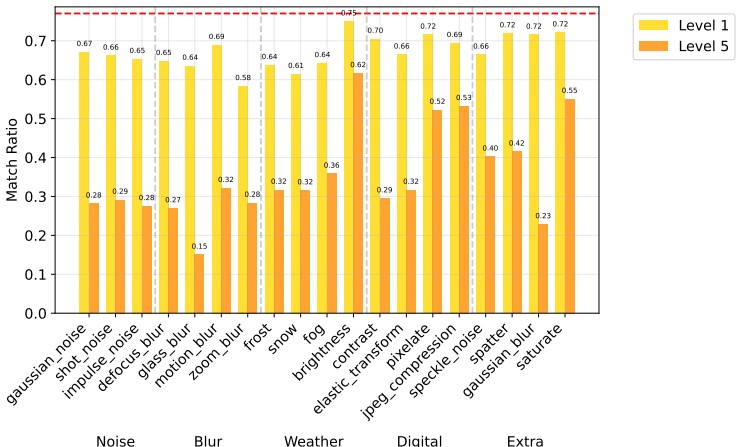

Figure 18: **Comparison of the top1 accuracy matches of the ImageNet-C categories.** As expected, the results of top1 accuracy matching correspond to those of top5 accuracy matching. As with the unaltered images, the values for top1 matching are lower than those for top5 matching.

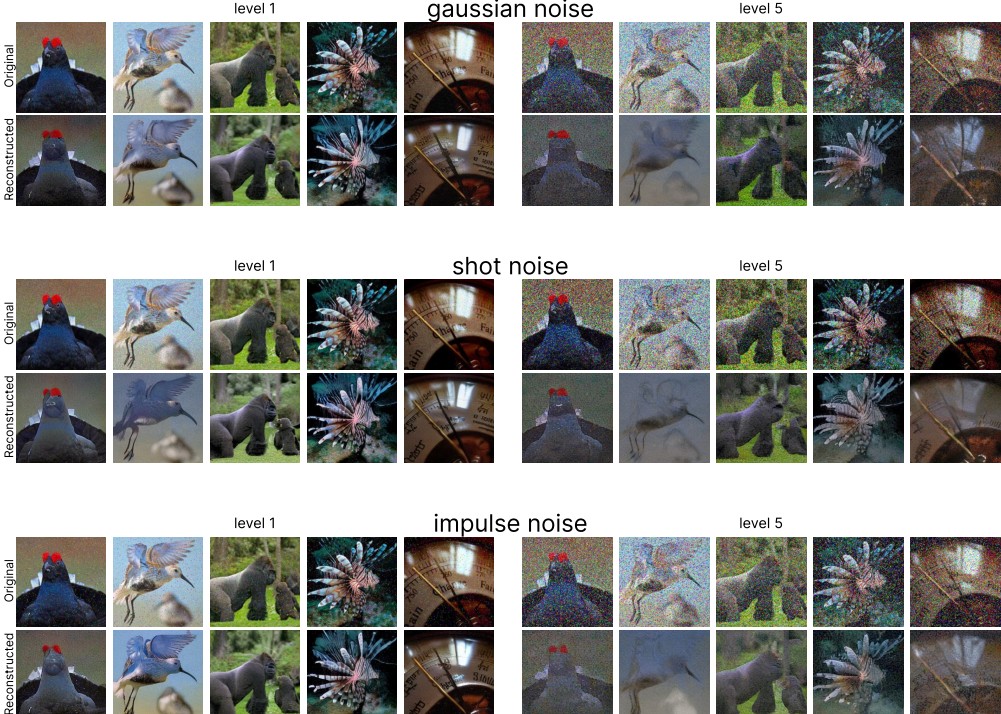

Figure 19: **Reconstructed corrupted images: Noise (gaussian noise, shot noise, and impulse noise).** In each category, the original corrupted images can be seen at the top and the reconstructed images below. On the left is level 1, i.e., weak corruption, and on the right is level 5, i.e., strong corruption. It can be seen that FeatInv is not very good at reconstructing detailed noise, and the images lose detail. In particular, the colors of the noise are lost.

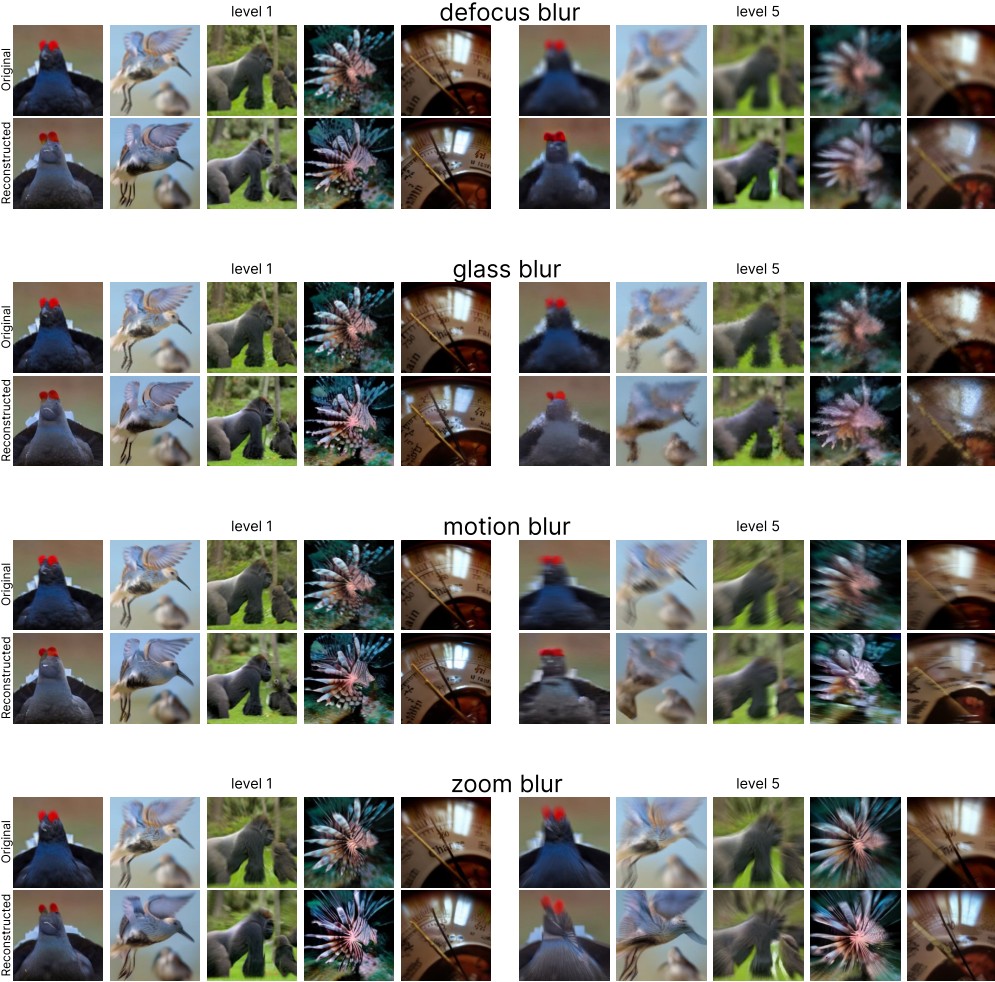

Figure 20: **Reconstructed corrupted images: Blur (defocus blur, glass blur, motion blur, and zoom blur).** 0 In each category, the original corrupted images are shown above and the reconstructed images below. On the left is level 1, i.e., weak corruption, and on the right is level 5, i.e., strong corruption. It can be seen that FeatInv is quite capable of reconstructing the images. Both levels 1 and 5 show good results.

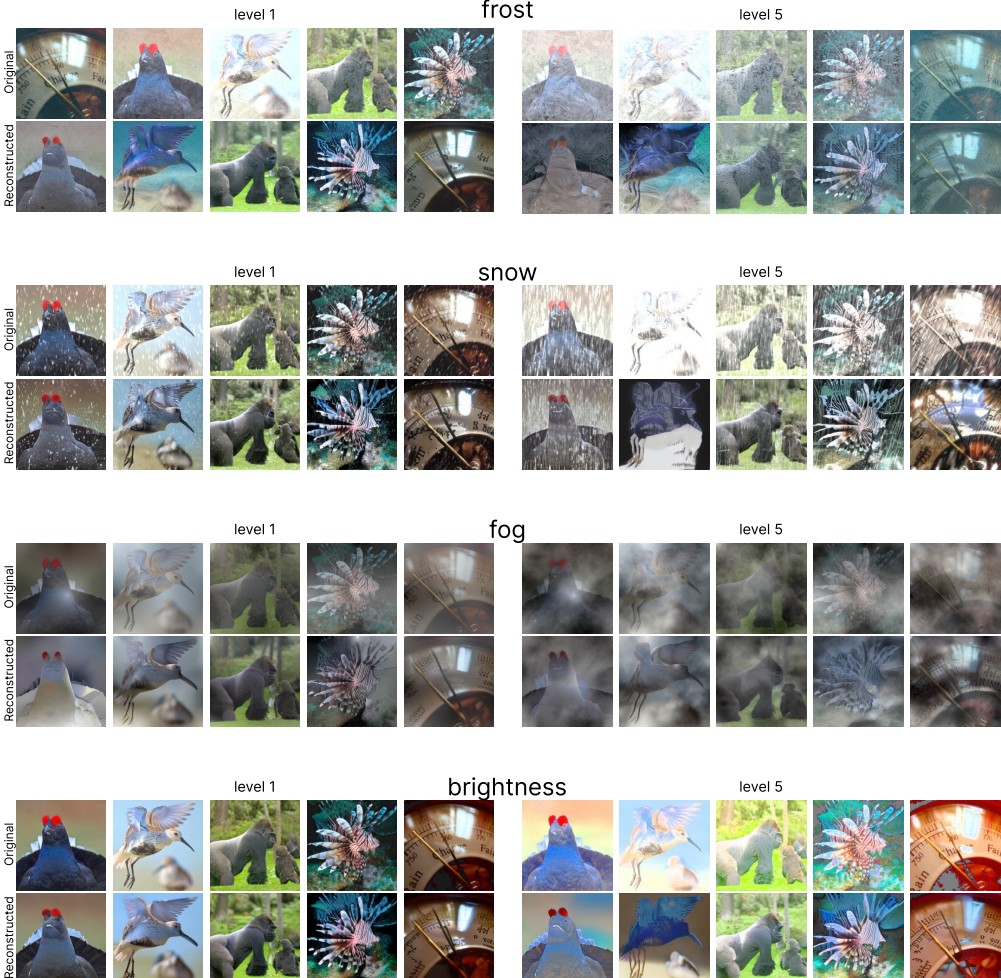

Figure 21: **Reconstructed corrupted images: Weather (frost, snow, fog, and brightness.)** In each category, the original corrupted images are shown above and the reconstructed images below. On the left is level 1, i.e., weak corruption, and on the right is level 5, i.e., severe corruption. It can be seen that FeatInv is quite capable of reconstructing the images. However, there are images where the reconstruction does not work as well, such as the second image, the flying bird. In all categories except fog, the reconstruction differs greatly from the original corrupted image.

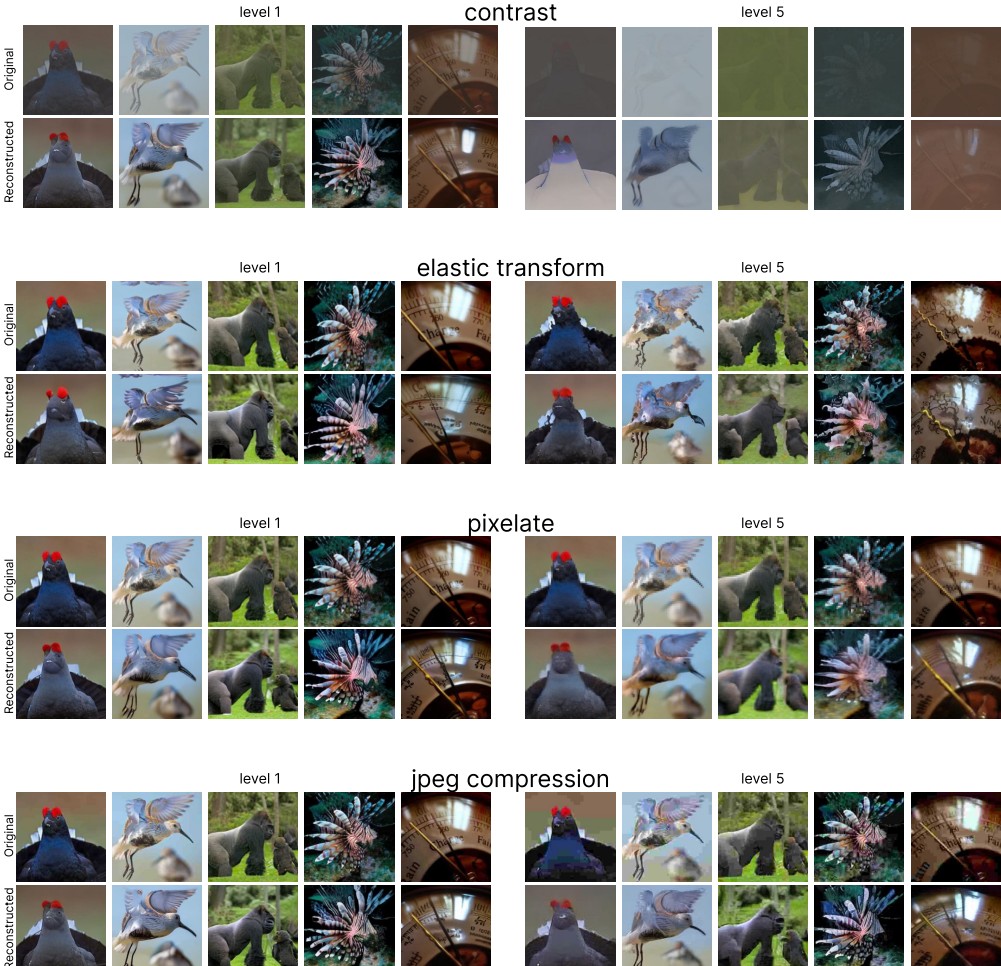

Figure 22: **Reconstructed corrupted images: Digital (contrast, elastic transform, pixelate, and jpeg compression.)** In each category, the original corrupted images are shown above and the reconstructed images below. On the left is level 1, i.e., weak corruption, and on the right is level 5, i.e., severe corruption. It can be seen that FeatInv is quite capable of reconstructing the images in the last three categories well. However, this does not work for contrast. Especially at level 5, the reconstructed images differ significantly from the original corrupted image. The main object is more emphasized than in the image with low contrast.

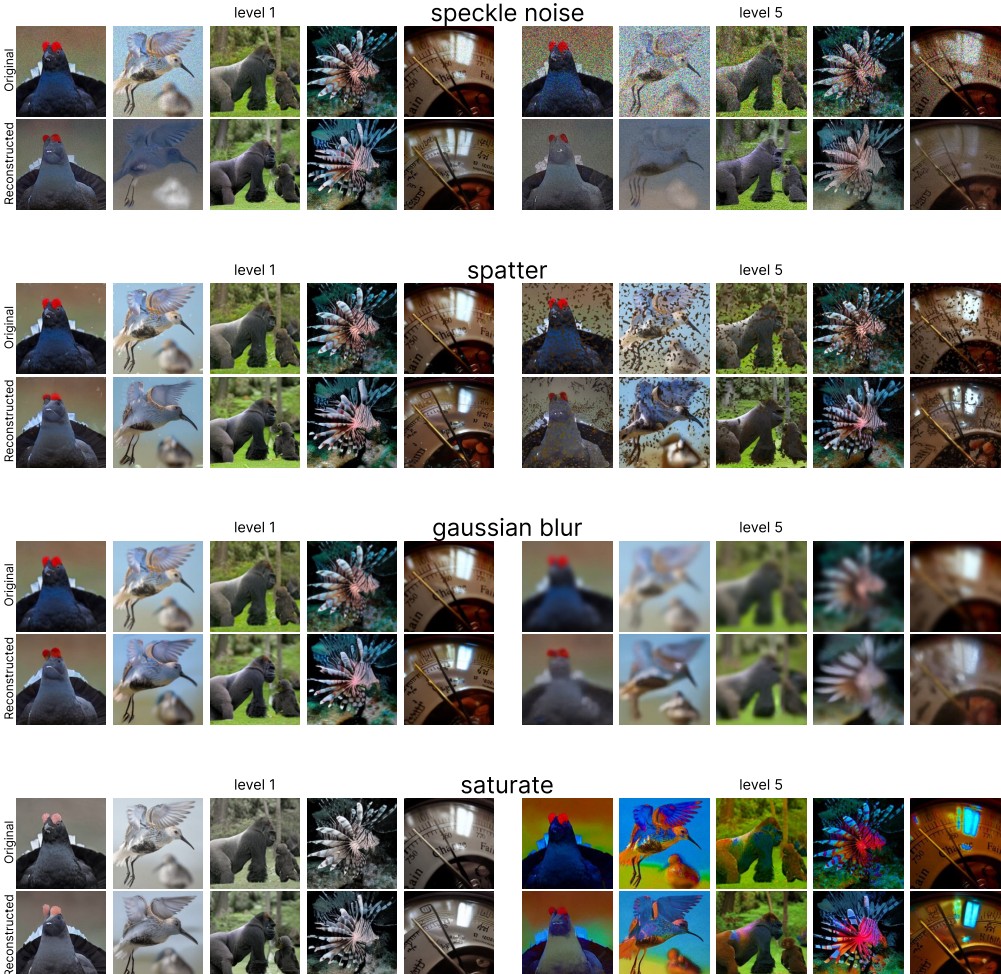

Figure 23: **Reconstructed corrupted images: Extra (speckle noise, spatter, gaussian blur and saturate.)** In each category, the original corrupted images are shown above and the reconstructed images below. On the left is level 1, i.e. weak corruption, and on the right is level 5, i.e. strong corruption. Except for the first category, speckle noise, FeatInv is able to reconstruct the images well. In the case of spatter, the dots are reconstructed, but some are located differently or are fewer in number. In the case of speckle noise, similar to the reconstructions in Figure 19, the noise is not well represented in the reconstructed images.

# D    Details on concept steering

For concept discovery, we rely on multi-dimensional concept discovery (MCD) (Vielhaben et al., 2023). For every feature vector $\phi$, MCD provides a concept decomposition $\phi = \sum_{i=1}^{n_c+1} \phi_i$, where $\phi_i$ is associated with the concept $i$ (of in total $n_c$ concepts), which represents a linear subspace of the feature space, and concept $n_c + 1$ corresponds to the orthogonal complement of the span of all concept subspaces. The latter is necessary to achieve a complete decomposition not explicitly captured through concept subspaces. For a given feature vector, one can now quantify the contribution $\phi_i$ from concept $i$ and visualize its magnitude $|\phi_i|_2$ across the entire feature map to obtain a spatially resolved concept activation map. One option to align such a coarse concept activation map with the input image is to use bilinear upsampling. This process often leads to rather diffuse concept activation maps. Even though we use MCD for demonstration, this alignment step is a common challenge for most concept-based attribution maps.

We used the learned mapping from feature space to input space to infer high-resolution concept visualizations. To this end, the component $\phi_i$ associated with concept $i$ was multiplied by 0.25 to attenuate them. In our experiments, decreasing the feature values worked better than increasing them. We speculate that increasing feature vector components can eventually result in feature vectors that exceed the magnitude of feature vectors seen during training of the FeatInv model.

To obtain higher-resolution representations, we used FeatInv to reconstruct five samples for each concept using both the original feature map and the concept-manipulated feature map. Using the same random seed for each pair ensured that the original and manipulated reconstructions were directly comparable, with differences attributable solely to the feature manipulation. For each pair, we computed the pixel-wise difference, and to produce a representative difference map for each concept, we took the median across the five resulting difference maps. This yielded a high-resolution (256×256) activation map that highlights the specific regions of the image affected by the manipulation.

## D.1    *FeatInv-Viz* visualization of other classes

To provide a closer look at *FeatInv-Viz*, we present visualizations of three concepts for the classes Great White Shark, Fly, African Elephant, and Water Ouzel. For the visualization we normalize the respective outputs of Algorithm 1 and threshold it below 0.33 as a binary mask to indicate unaffected regions of the image.

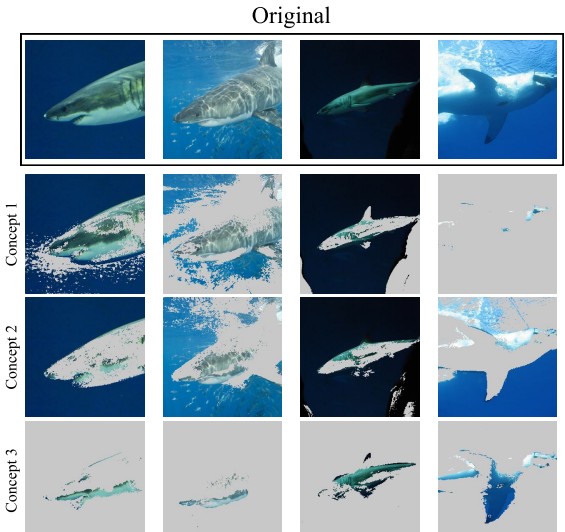

Figure 24: ***FeatInv-Viz* visualization of Great White Shark concepts** identified within ConvNeXt's feature space of the Great White Shark class, which can be associated with shark/water, water and shark. Concept 1 and 2 both seem to show water while concept 1 also highlights parts of the shark.

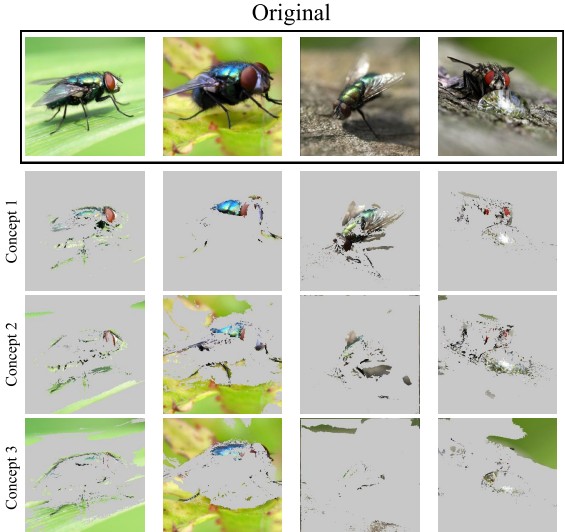

Figure 25: *FeatInv-Viz* **visualization of Fly concepts** identified within ConvNeXt's feature space of the Fly class, which can be associated with fly's body, outline/part of fly/leaves and leaves/green. Interestingly, Concept 3 focuses only on the supposedly green background in (see images 3 and 4). The rest of the background does not appear to be part of the concept.

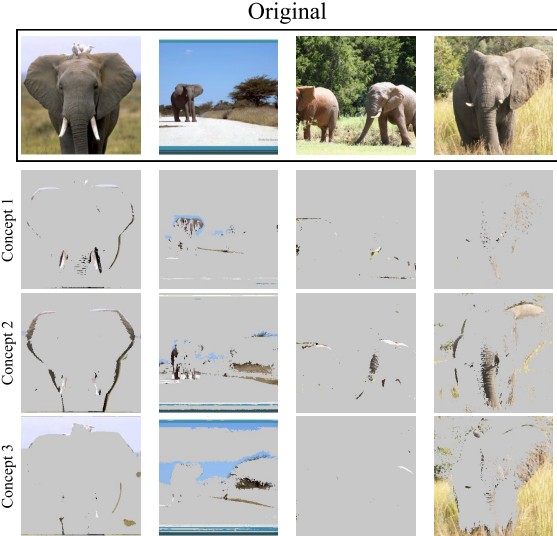

Figure 26: *FeatInv-Viz* **visualization of African Elephant concepts** identified within ConvNeXt's feature space of the African Elephant class, which can be associated with tusks, outline/part of elephant and grass/sky. Unlike with the Great White Shark or Fly class, only the outlines of the elephant concept can be seen here.

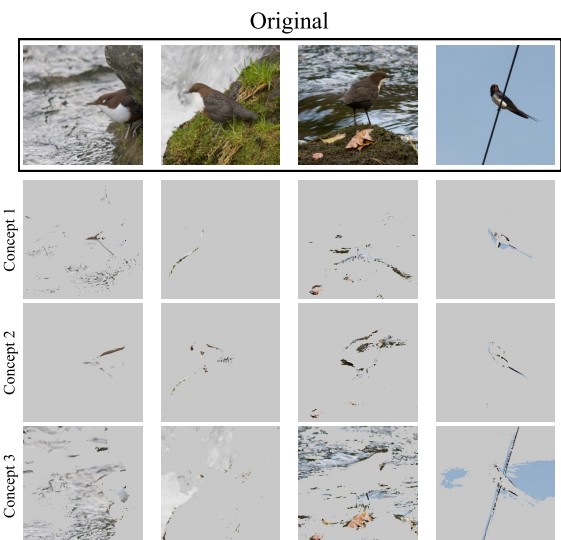

Figure 27: *FeatInv-Viz* **visualization of Water Ouzel concepts** identified within ConvNeXt's feature space of the Water Ouzel class, which can be associated with water/sky boundary, outline/part of bird and water/sky. As in the concepts of the African Elephant, only an outline of the bird can be seen in these representations.

