# OpenReview forum: "FeatInv: Spatially resolved mapping from feature space to input space using conditional diffusion models"
_TMLR — Accepted by TMLR_

### Review · Reviewer_qsaR · 2025-09-13

**Summary Of Contributions:**

## Summary
This paper introduces FeatInv, inverting feature representations back to input space by conditioning a pretrained diffusion model (ControlNet-style) on spatially resolved feature maps. While feature inversion for explainability has been studied extensively in the deep learning community, this work provides an empirical extension by leveraging ControlNet to reconstruct high-resolution, visually realistic inputs. The work also demonstrates applications such as concept steering and compositional feature visualization.


## Strengths
I believe this work will be of interest to the community.
1. While feature inversion has been extensively studied, this paper leverages diffusion models (ControlNet-style conditioning) to achieve higher-fidelity and probabilistic reconstructions, offering a clear improvement over prior deterministic approaches.
2. This paper showcases interesting applications, including concept steering visualization and compositional feature analysis, highlighting the broader utility of the method.

## Weaknesses
1. **Incremental advances in feature inversion**. Prior work [1-3] already explored feature inversion with generative models. The main difference here is conditioning diffusion models on unpooled feature maps.
2. **Justification of feature map choice**. The argument regarding **pooled** and **unpooled** representations not fully justify the design decision. The paper claims that conditioning on feature maps before pooling leads to better reconstructions. However, this naturally raises the question: why not condition on even earlier layers, which may contain richer spatial detail? The work would benefit from a clearer rationale for focusing specifically on the last pre-pooling feature map, rather than earlier alternatives.
3. **Faithfulness of explanations**. While the reconstructions are visually compelling, the work does not establish whether they are faithful explanations of the underlying model’s reasoning [4]. I feel reconstruction resolution is not a major concern in explainability. Without guarantees or evaluations of faithfulness, it remains unclear to what extent the reconstructed images truly reflect the model’s decision process rather than artifacts introduced by the generative prior.
4. **Experiments**. It would be valuable to examine how the method explains perturbed samples, such as adversarially perturbed inputs, corrupted images, or other distribution shifts.



References:
[1] Alexey Dosovitskiy, Thomas Brox, Generating Images with Perceptual Similarity Metrics based on Deep Networks
[2] Florian Bordes, Randall Balestriero, Pascal Vincent, High Fidelity Visualization of What Your Self-Supervised Representation Knows About
[3] Robin Rombach, Patrick Esser, Björn Ommer, Making Sense of CNNs: Interpreting Deep Representations & Their Invariances with INNs
[4] Chih-Kuan Yeh, Cheng-Yu Hsieh, Arun Sai Suggala, David I. Inouye, Pradeep Ravikumar, On the (In)fidelity and Sensitivity for Explanations

**Additional Comments:**

NA

**Audience:**

Yes

**Audience Explanation:**

I believe this work will be of interest to the community.

**Claims And Evidence:**

Yes

**Claims Explanation:**

NA

**Requested Changes:**

## Requested revisions.
1. I would like to suggest the authors re-state the claims in the contributions. In particular, regarding the positioning of this work regarding how it advances the feature inversion research line.
2. While the method achieves good  reconstruction fidelity, this alone does not establish its value for explainable AI. High-fidelity reconstructions demonstrate that the model can generate realistic images aligned with feature representations, but they do not by themselves clarify how those features contribute to the model’s decision-making. A stronger connection between reconstruction fidelity and explanatory faithfulness is needed to justify the interpretability claims.
3. Regarding the argument the pooled versus unpooled. The high reconstruction quality can't justify this work. We need the good causal connection between explanations and decisions instead of the reconstructed quality.
4. Experiments. Would it be possible to assess how adversarial samples, corrupted samples, OOD affect this framework?

---

> ### Author Response · Authors · 2025-10-13
> **Clarifying FeatInv’s Faithfulness, Feature Map Choice, and New Experimental Results**
>
> - **Incremental advances in feature inversion:** We acknowledge the reviewer's perception that nature of the conditional information seems like a minor detail. We believe that our experiments convincingly demonstrate the shortcomings of pooled representations if faithful reconstructions are required. This applies even more to conditioning on feature maps of intermediate feature layers where the pooled representation does not carry a proper meaning.
>
> - **Justification of feature map choice:** We thank the reviewer for this remark. We argued that the framework was applicable to any feature layer but only presented results for the last. We believe that the final feature map is special in the sense that the final feature map and the final pooling before the classifier provide the vectors that lead to the model's classification decision. We deliberately chose the last layer to visualize the data on which a model bases its decision. However, FeatInv is also capable of being conditioned on earlier layers. We conducted additional experiments, see  Section 4.1: Conditioning on intermediate feature layers, to demonstrate the feasibility of conditioning on intermediate feature maps. We believe that the choice of the layer to condition on should be dictated by the downstream application for such a visualization. If the goal is to provide insights for layer h, also FeatInv should be trained using layer h as conditional input.
>
> - **Faithfulness of explanations:** To put the discussion on firm ground, we deem it worthwhile to provide a definition of faithfulness. We consider a feature inversion model as faithful if it yields an input reconstruction that yields good alignment with the full feature map and good alignment with the original model output. We clarified this in Section 4.1: Reconstruction Quality. We believe that the FeatInv approach should be ultimately assessed through metrics that quantify the quality of the mapping using the metrics put forward in Section 4.1. However, as part of the experiment to assess the applicability to intermediate feature layers, we also evaluated the reconstruction quality for other layers the model was not explicitly conditioned on, and found also good alignment. The most probable explanation for this behavior is that FeatInv conditioned on any kind of feature map input yields an reconstructed input in input space that is close to the original sample and therefore also matches activations extracted from layers FeatInv was not conditioned on. We argue that these results support the faithfulness of the model inversion process. We stress that FeatInv by itself does not provide any explanation but can be used to improve feature visualizations in XAI methods.
>
> - **Experiments:** We appreciate the reviewer's question for additional robustness evaluations. With feature map manipulation such as cutmix and mixup, we have conducted some experiments in this direction at the feature map level. Apart from OOD (which we moved from the supplementary material to Section 4.2 and added cosine similarity as well as FID score), we have not conducted any experiments at the input image level. We included further experiments:
>
>   - Reconstruction on adversarial samples (see Section 4.2: Evaluation with adversarial images)
>
>   - Reconstruction on corrupted images (see Section 4.2: Evaluation with corrupted images, Supp Mat. C)
>
>   These experiments support OOD evaluation and feature manipulation experiments included in the original draft and showcase the robustness of our approach.
>
> - **Feature inversion claims in the contributions:** We framed the first contribution more clearly by adding a "spatially resolved" qualifier. Secondly, we extended the contributions by third point to account for the comprehensive robustness evaluations added at the request of the reviewer.
>
> - **How features contribute to model's decision making:** From our point of view FeatInv is faithful for the task it was built for (see remarks on faithfulness above) and this property mainly builds on reconstruction quality. Its goal is not to elucidate the internal decision making process, but it can be used to provide hints on the latter, as to understand the nature of the feature space at different layers, which could then be connected across layers through circuits or other mechanistic XAI constructions.
>
> - **Motivation pooled vs. Unpooled:** Leveraging faithfulness as defined above, we establish a close connection between feature space and input space. This allows to manipulate feature space and investigate corresponding changes in the input space, which is the core idea that stands behind FeatInvViz. Without faithfulness, i.e., when conditioning on pooled feature maps this is not possible.

---

### Review · Reviewer_4GxC · 2025-09-21

**Summary Of Contributions:**

This paper introduces FeatInv, a novel method for mapping the internal feature representation of a deep neural network to a high-fidelity image. The process is inversion in the sense that the generated image should produce a similar feature representation to the original image. The core contribution is the use of a pretrained conditional diffusion model to learn this mapping. Unlike previous methods that often rely on deterministic or approximate approaches, FeatInv leverages the generative power of diffusion models to produce high-fidelity, spatially resolved images conditioned on specific feature maps. The authors explore the potential of this idea and validate the method's effectiveness through numerical experiments. Furthermore, the paper illustrates practical applications, such as visualizing concept steering and exploring the composite nature of feature representations, thereby offering a powerful new tool for network interpretability.

**Audience:**

Yes

**Audience Explanation:**

This work will be of high interest to researchers and practitioners in machine learning, particularly those focused on computer vision and model interpretability. The problem of understanding what neural networks learn is a fundamental and persistent challenge in the field. This paper presents a clear, well-motivated, and technically sound approach.

**Broader Impact Concerns:**

None.

**Claims And Evidence:**

Yes

**Claims Explanation:**

The claims are backed by numerical experiments.

**Requested Changes:**

The paper is well-written and the core methodology seems sound. I appreciate the fact that the proposed idea is simple, yet well-explored.

In the "Architecture and training procedure" section, the final sentence is grammatically incorrect.

In the first sentence of the "Conditional input encoder" section, the word "import" should be corrected to "important".

---

> ### Author Response · Authors · 2025-10-13
> **Correction of Language Issues**
>
> - **Overall:** we thank the reviewer for the positive assessment  of our work and also the appreciation of the broad applicability of the proposed method.
>
> - **Grammatical errors:** We thank the reviewer for the careful reading of our manuscript and for highlighting several grammatical and typographical errors. We fixed these in the revised version of the manuscript.

---

### Review · Reviewer_d8wL · 2025-10-06

**Summary Of Contributions:**

This paper proposes FeatInv, a conditional diffusion model for reconstructing input image from feature representation of pre-trained vision encoders. The goal is to obtain high-fidelity image reconstructions whose encoded features align with those of the original input. The main idea is to finetune a pretrained diffusion model in a ControlNet fashion with conditions from pre-pooling features.

**Summary of contributions**
- Implements FeatInv, a ControlNet variant that can reconstruct natural images from their pre-pooling feature maps of vision encoders.
- Produces visually coherent and realistic reconstructions, suggesting that diffusion-based inversion is feasible for feature-space visualization.
- Presents qualitative applications of FeatInv for visualization of concept direction and concepts composition in the feature space.

**Strengths**
- Provide both quantitative and qualitative results to show that pre-pooling features preserve sufficient local structure for high-quality reconstruction.
- Consistent reconstruction and classification alignment across three diverse backbones.

**Weaknesses**
- The main methodological difference from RCDM (Bordes et al. 2022) is limited. The primary changes include the use of pre-pooling feature instead of post-pooling feature, and the use of ControlNet for conditioning.
- To the best of my knowledge, the value of FeatInv is unclear beyond visualization.
	- FeatInv does not discover, analyze, or interpret internal concepts. It assumes known concept vectors (e.g., from MCD) and visualizes their attenuation in input space.
	- Feature composition results do not seem realistic.
- The authors did not discuss the choice of pre-trained diffusion model. Does the diffusion model need to be trained on the same distribution as the vision encoder? More importantly, to what extent the reconstruction is biased by the pre-trained diffusion model?

**Audience:**

No

**Audience Explanation:**

While FeatInv confirms that diffusion-based inversion of feature maps is feasible and produces visually realistic images, its scientific contribution is limited. The best use case might be visualization tool for feature space but it does not provide new insights into model representations.

**Claims And Evidence:**

Yes

**Claims Explanation:**

The qualitative and quantitative results convincingly demonstrate that conditioning on spatial (pre-pooling) features leads to higher reconstruction fidelity. The concept-steering and feature-composition results are mostly qualitative demonstrations.

**Requested Changes:**

- Beyond the use of pre-pooling features and ControlNet conditioning, what fundamental problem or capability does FeatInv address that RCDM cannot?
- Could you discuss how the pre-trained diffusion model prior might bias the reconstruction? For instance, what happens if the diffusion model is trained on a dataset different from the encoder’s (e.g., FFHQ vs. ImageNet)?
- Can FeatInv support quantitative interpretability, debugging, or representation analysis? Could you discuss what additional insights it can provide beyond rendering visually plausible images?

---

> ### Author Response · Authors · 2025-10-13
> **Clarifying FeatInv’s Relevance and Technical Contributions [1/2]**
>
> - **Main innovation:** We agree with the reviewer's assessment and did also clearly acknowledge in the orginal submission that the main extension from the procedural point of view was the extension from unpooled towards pooled feature maps as conditional information. Nevertheless, implementing this through a ControlNet-style design and assessing its impact is a non-trivial step. We believe that a subjective assessment of the degree novelty of the proposed approach does not align with TMLR's focus on technical correctness.
>
> - **Lack of interest in the TMLR audience:** We believe that FeatInv is a quite versatile tool that could be relevant for different sub-communities (XAI, representational analysis to name just two). The two other reviewer did not share the pessimistic view on a lack of interest in the community.
>
> - **Value beyond visualization:** As stated correctly by the reviewer, FeatInv should primarily be seen as a visualization tool. However, this visualization step is non-trivial and can provide insights (c.f. below) and we provide first indications on how it could be applied for example to concept-based XAI pipelines.
>
> - **FeatInv does not discover, analyze or interpret concepts:**  As stated in the previous response, FeatInv is a visualization tool and therefore does not aim to discover concepts. We believe that the ability to trace back structures probed through manipulations in feature space to finegrained localized structures in input space is a non-trivial finding, which clearly extends beyond the commonly used state-of-the-art in the field such as upsampling. The inability to assess the meaning of discovered (unsupervised) concepts or to trace back relevant regions for discovered prototypes is a major bottleneck in many XAI methods, which could be enhanced through FeatInv as modular component. Therefore, we believe that FeatInv in fact helps to interpret concepts. However, we also believe that FeatInv has further applications beyond XAI and therefore rather presented it as a general mapping tool between feature space and input space.
>
> - **Feature composition not realistic:** We don't believe there is a proper ground truth to assess the realism of interpolated samples.
>
> - **Impact of pretrained diffusion model:** We thank the reviewer for this remark and fully agree that the impact of the pretrained diffusion model is one of the few design choices that we have not investigated. First, this is a proof-of-principle study to demonstrate the feasibility and the gains in faithfulness achieved through conditioning on spatially resolved feature maps. Second, we wanted to avoid training the pretrained diffusion model ourselves to avoid issues due to suboptimal (pre)training, which left us with MiniSD as to the best of our knowledge only pretrained diffusion model at low resolution of around 224x224 pixels (chosen for computation reasons and for later application with downstream classifiers operating at 224x224). Third, we believe that our OOD evaluation on the BAID dataset provides some insights on the robustness of our approach, even in the case where not only the pretrained diffusion model but also the ControlNet is confronted with OOD inputs. However, this experiment cannot disentangle effects between the mismatches of data distributions used during pretraining and during ControlNet training but don't think this represents a serious limitation given the scope of the submission.
>
> - **Added value beyond RCDM:** We believe that our experiments convincingly demonstrate that conditioning on pooled feature maps (a la RCDM) leads to input reconstructions that capture the semantic content but not precise details such as the spatial composition of the image. We therefore find it questionable to rely on such a mapping for purposes where a faithful mapping is required (which includes XAI application as well as robustness/representational stability studies). One might argue that for the final hidden layer also the pooled representation is an appropriate object to condition on, however, this is certainly not the case for intermediate feature layers, which are typically not used after pooling. Extended experiments requested by Reviewer qsaR demonstrate that FeatInv is also applicable to these cases, see Section 4.1: Conditioning intermediate feature layers and Supp Mat. B. To summarize, while the nature of the conditional information might seem like a minor detail, we believe it has quite important implications for downstream applications.

---

> > ### Author Response · Authors · 2025-10-13
> > **Clarifying FeatInv’s Relevance and Technical Contributions [2/2]**
> >
> > - **FeatInv supporting quantitative interpretability, debugging, or representation analysis:** FeatInvViz allows finegrained visualizations of substructures in feature space mapped back to input space. This includes all three aspects mentioned above (quantitative interpretability: mapping out input regions that align with concepts under consideration with unprecedented precise regions; debugging: improved feature visualization will also improve clever hans detection; representation analysis: the method allows to study the impact of manipulations in feature space directly in input space, we provide some initial experiments on mixup/cutmix applied in feature space). We are not convinced that "visually plausible" is a fair assessment of the produced samples, as these inputs produce feature maps that are very close to those used for conditioning and therefore correspond to natural images as perceived by the model.

---

### Decision · Action_Editor_oDku · 2025-11-10

**Recommendation:** Accept with minor revision

**Additional Comments:**

Major comment: As noted by the reviewers, prior work has already investigated feature inversion using various generative models. The authors should include experimental comparisons between the proposed FeatInv and representative existing methods (even though some of the methods may not be applicable to various architectures), or provide justification why such comparison is not missing.

Minor comment:
1. page 6: two is in the fifth line "this objective is is implemented"
2. page 8: 10.000 should be 10,000

**Audience:**

Yes

**Audience Explanation:**

The result would be of interest to researchers working on interpretability, feature visualization, and model inversion.

**Claims And Evidence:**

Yes

**Claims Explanation:**

This paper introduces FeatInv, a conditional diffusion model designed to reconstruct input images from hidden-layer feature representations. The key idea is to fine-tune a pre-trained diffusion model, conditioning it on hidden-layer features to faithfully recover the corresponding input images.

To evaluate FeatInv’s performance, the authors conduct experiments on several backbone architectures, including ResNet-50, ConvNeXt, and SwinV2, demonstrating that the reconstructed images can effectively capture the visual and semantic content encoded in the features through two measures (1) cosine similairty between the features of the orinal and reconstructed images, and (2) classifiction accuracies for the generated images.

In addition, the paper showcases some potential applications of the proposed FeatInv, such as concept steering visualization and compositional feature analysis.

---

> ### Author Response · Authors · 2025-12-05
> **Extended Experimental Comparison with RCDM**
>
> We sincerely thank the Action Editor for the constructive assessment and positive recommendation. In accordance with the request for additional experimental comparison, we have extended our experiments to include our closest competitor, RCDM (Bordes et al., 2022), with results provided in Appendix Section A.5. Specifically, we now include evaluations using both the publicly available pretrained supervised ResNet-50 RCDM model as well as an independently trained RCDM model based on ConvNeXt. These additional experiments substantiate our claims and further highlight the advantages of FeatInv in terms of reconstruction fidelity.